# TRPV1 regulates excitatory innervation of OLM neurons in the hippocampus

Joaquin I. Hurtado-Zavala[1], Binu Ramachandran[1], Saheeb Ahmed[1,2], Rashi Halder[3], Christiane Bolleyer[1], Ankit Awasthi[1], Markus A. Stahlberg[1], Robin J. Wagener[4,5], Kristin Anderson[6], Ryan M. Drenan[7], Henry A. Lester[8], Julie M. Miwa[6], Jochen F. Staiger[4], Andre Fischer[3] & Camin Dean[1]

TRPV1 is an ion channel activated by heat and pungent agents including capsaicin, and has been extensively studied in nociception of sensory neurons. However, the location and function of TRPV1 in the hippocampus is debated. We found that TRPV1 is expressed in oriens-lacunosum-moleculare (OLM) interneurons in the hippocampus, and promotes excitatory innervation. TRPV1 knockout mice have reduced glutamatergic innervation of OLM neurons. When activated by capsaicin, TRPV1 recruits more glutamatergic, but not GABAergic, terminals to OLM neurons in vitro. When TRPV1 is blocked, glutamatergic input to OLM neurons is dramatically reduced. Heterologous expression of TRPV1 also increases excitatory innervation. Moreover, TRPV1 knockouts have reduced Schaffer collateral LTP, which is rescued by activating OLM neurons with nicotine—via α2β2-containing nicotinic receptors—to bypass innervation defects. Our results reveal a synaptogenic function of TRPV1 in a specific interneuron population in the hippocampus, where it is important for gating hippocampal plasticity.

[1] Trans-synaptic Signaling Group, European Neuroscience Institute, Grisebachstrasse 5, Goettingen 37077, Germany. [2] Department of Diagnostic and Interventional Radiology, University Medical Center Göttingen, Robert Koch Strasse 40, Göttingen 37075, Germany. [3] Department of Epigenetics and Systems Medicine in Neurodegenerative Disease, German Center for Neurodegenerative Disease, University Medical Center Goettingen, Grisebachstrasse 5, Goettingen 37077, Germany. [4] Institute for Neuroanatomy, University Medical Center Goettingen, Kreuzbergring 36, Goettingen 37075, Germany. [5] Department of Basic Neurosciences, Faculty of Medicine, University of Geneva, Geneva CH-1211, Switzerland. [6] Biological Sciences Department, Lehigh University, Bethlehem, Pennsylvania 18015, USA. [7] Department of Medicinal Chemistry and Molecular Pharmacology, Purdue University, West Lafayette, Indiana 47907, USA. [8] Division of Biology and Biological Engineering, California Institute of Technology, 1200 East California Boulevard, Pasadena, California 91125, USA. Correspondence and requests for materials should be addressed to C.D. (email: c.dean@eni-g.de).

The transient receptor potential vanilloid 1 (TRPV1) channel is a non-selective cation channel activated by heat, acidic conditions and naturally occurring exogenous molecules including capsaicin, the pungent ingredient in chilli peppers[1–6]. TRPV1 was first discovered and characterized in the peripheral nervous system, where it plays a prominent role in nociception[1].

The presence of TRPV1 in the brain, and especially in the hippocampus, has been debated for many years. TRPV1 was initially reported to be absent in the brain[1]. Later studies, however, reported broad expression of TRPV1 in the brain including the hippocampus[7–9], where it was observed in cell bodies of pyramidal neurons throughout the CA1 and CA3 regions and the dentate gyrus of the hippocampal formation[9]. However, more recent studies have challenged the notion that TRPV1 is broadly expressed in the hippocampus. Only a small population of cells in the hippocampus was found to express TRPV1 in a reporter mouse line in which lacZ expression is driven by the TRPV1 promoter[10]. The few TRPV1-positive cells in the hippocampus co-expressed reelin, and were hypothesized to correspond to Cajal–Retzius cells, which die during development, and TRPV1 was not detected in the adult hippocampus by RT–PCR[10].

However, TRPV1 has been implicated in synaptic plasticity in the hippocampus since the observation that TRPV1 knockouts have decreased long-term potentiation (LTP) of synaptic strength in the Schaffer collateral pathway compared to wild type (WT)[11]. Addition of exogenous capsaicin to hippocampal slices also has been reported to increase LTP[12,13], further suggesting that functional TRPV1 is present in the adult hippocampus, and that TRPV1, if activated by endogenous ligands, may affect LTP.

Subsequent studies resulted in disparate findings regarding the role of TRPV1 in hippocampal synaptic plasticity. One model proposed that high-frequency stimulation of the Schaffer collateral pathway activates presynaptic TRPV1 on pyramidal neurons, which persistently reduces glutamate release onto interneurons[14]. Thus TRPV1 knockouts would have higher inhibition and a subsequent reduction in Schaffer collateral LTP. By contrast, in the perforant pathway it was proposed that high-frequency stimulation activates post-synaptic TRPV1 on pyramidal neurons, which induces endocytosis of AMPA receptors, and would result in an increase in LTP in TRPV1 knockouts[15]. However, TRPV1 was not localized in these studies, and these models are both at odds with reports of restricted expression of TRPV1 in the adult hippocampus[10]. Thus, both the presence of TRPV1 in the hippocampus—whether broadly expressed, sparsely expressed, or not at all—and how TRPV1 affects hippocampal plasticity and function, is still unclear.

Most previous studies describing a function of TRPV1 in synaptic plasticity have localized TRPV1 to excitatory neurons[7,9,14], which comprise the principle cells of the glutamatergic tri-synaptic circuit in the hippocampus. However, TRPV1 has also recently been reported at post-synaptic sites in inhibitory neurons[16]. Although inhibitory interneurons represent only ∼10% of the neurons in the hippocampus, it has recently become clear that diverse populations of inhibitory interneurons—where at least 21 different types have been described in the CA1 region alone[17]—are key players in the control of principal cells in the hippocampus to modulate synaptic plasticity[18].

Here we discovered, using TRPV1 knockouts to validate quantitative PCR (qPCR) primers and TRPV1 antibodies, that TRPV1 is expressed in oriens-lacunosum-moleculare (OLM) (oriens-lacunosum-moleculare) interneurons in the hippocampus. TRPV1-expressing OLM neurons have profuse excitatory innervation, which is significantly reduced in TRPV1 knockouts. Upon stimulation with capsaicin, excitatory input to these OLM neurons was increased, and this effect was blocked by a TRPV1-specific

antagonist and abolished in TRPV1 knockouts. Moreover, heterologous expression of TRPV1 in neurons that do not normally express it, increased excitatory, but not inhibitory, input to these neurons.

We further found that the reduced Schaffer collateral LTP observed in hippocampal slices from TRPV1 knockouts is caused by the decrease in excitatory innervation of OLM cells: LTP in TRPV1 knockouts was rescued by specifically activating OLM neurons with nicotine—via the α2 receptor—to bypass innervation defects. Together our data show specific expression of TRPV1 within a sub-population of hippocampal neurons and reveal a novel function of TRPV1 channels in the hippocampus in regulating excitatory innervation, which explains previously reported deficits in synaptic plasticity observed in TRPV1 knockouts.

## Results

**TRPV1 is expressed in the hippocampus**. To test if TRPV1 is present in the hippocampus, we first examined TRPV1 mRNA levels in adult WT and TRPV1 knockout animals by qPCR. We used dorsal root ganglia (DRG) as a positive control for detection of TRPV1, since DRG are reported to have high levels of TRPV1 expression[19,20]. As expected, the sequence that is deleted in the TRPV1 knockout mouse corresponding to exon 13 and encoding the putative TRPV1 pore region (TRPV1 PR), was present in WT DRG but absent in TRPV1 knockouts (Fig. 1a). Importantly, TRPV1 mRNA in WT mice was also detected in the hippocampus, albeit at much lower levels than in DRG.

In addition to the sequence encoding the pore region, we also used primers to detect N- and C-terminal sequences of TRPV1–which have been used to generate many commercially available TRPV1-specific antibodies–and the VR.5′ sv splice isoform as a positive control, which has been reported to be present at higher levels in hippocampus than in DRG[21]. We found all three sequences present in the hippocampus of WT mice, where VR.5′ sv levels were higher in hippocampus than in DRG, as expected. Surprisingly, however, mRNA sequences corresponding to the TRPV1 N-terminus, C-terminus and the VR.5′ sv splice isoform of TRPV1 remained in the TRPV1 knockouts, in both DRG and hippocampus (Fig. 1b).

To determine if the TRPV1 channel is expressed in brain and hippocampus at the protein level, we performed western blots using an N-terminus-specific TRPV1 antibody in whole brain and hippocampal homogenates. A band of ∼90 kDa, corresponding to TRPV1, was detected in WT homogenates of whole brain and hippocampus, and was absent in TRPV1 knockouts (Fig. 1c). The C-terminal antibodies that we tested by western blot, did not yield a clear band of the correct size (Supplementary Fig. 1).

To further test if functional TRPV1 is present in the hippocampus, we examined calcium responses of hippocampal neurons to capsaicin, which specifically activates the TRPV1 channel. TRPV1 is permeable to cations including $Ca^{2+}$[1]. Monitoring $Ca^{2+}$ influx upon TRPV1 activation therefore allows us to determine if TRPV1 is expressed in hippocampal neurons as an active, functional channel. We performed time-lapse imaging of Fluo-4-loaded hippocampal neurons during 40 Hz field stimulation (as a positive control of calcium influx in response to depolarization) followed by 1 μM capsaicin stimulation. A small number of hippocampal neurons in vitro that responded to field stimulation also responded to capsaicin treatment (Fig. 1d). This response remained in the presence of TTX and CNQX, but was blocked by the TRPV1 antagonist SB-366791 (Supplementary Fig. 2). In addition, TRPV1 knockout neurons did not respond to capsaicin, indicating that the capsaicin response in WT neurons occurs via the TRPV1 channel. None of

the neurons analysed responded to application of the vehicle dimethylsulphoxide (DMSO) used as a negative control. Although the peak of the fluorescence signal corresponding to $Ca^{2+}$ influx during electrical stimulation and capsaicin treatment was variable among different neurons, the magnitude of the

capsaicin response was usually ∼ 50% of that evoked by electrical stimulation. The decay of $Ca^{2+}$ signal in response to capsaicin was also more gradual than in response to field stimulation, and in some cases the capsaicin-mediated $Ca^{2+}$ response did not decay completely to baseline before termination of recording

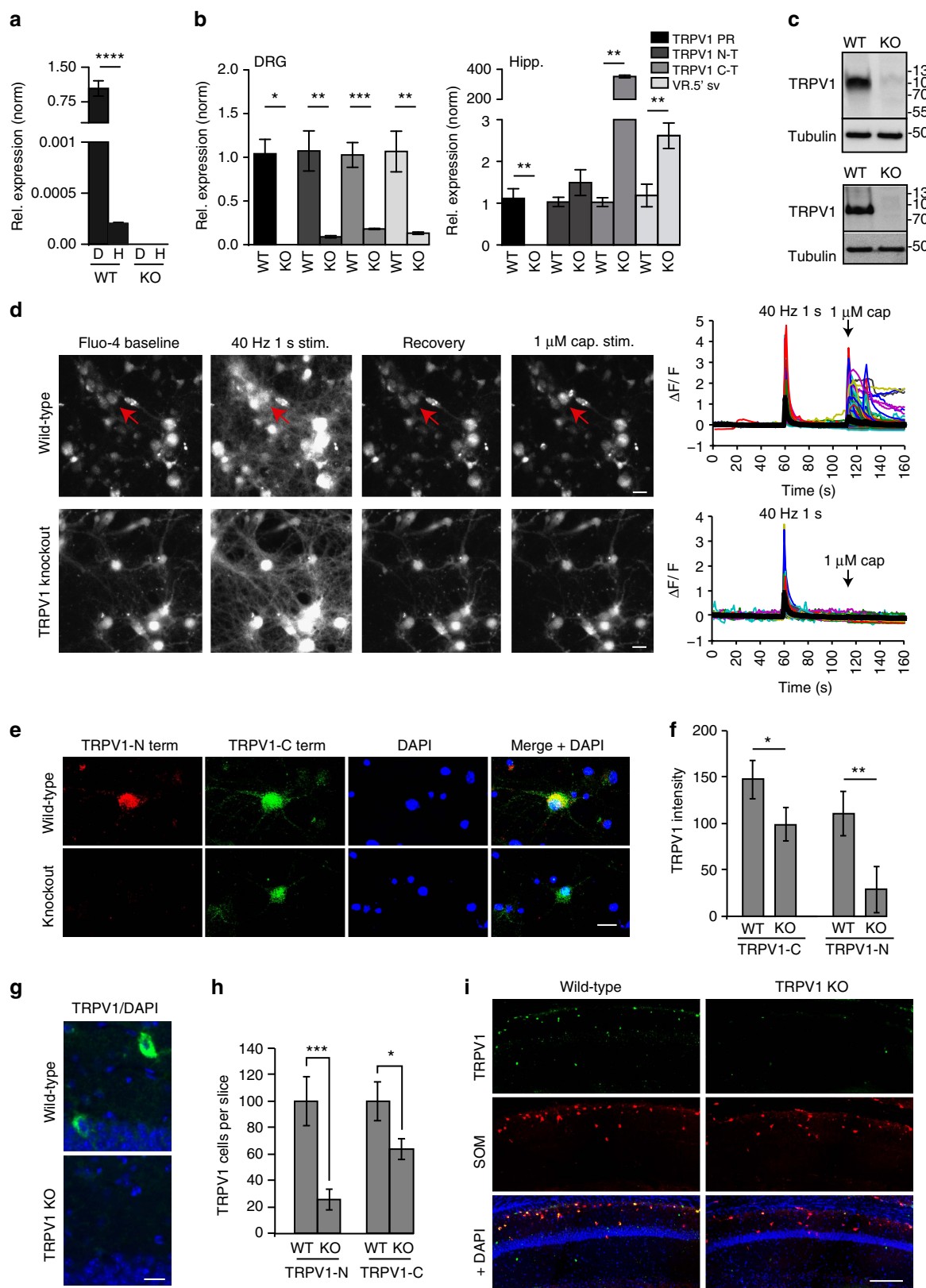

(Fig. 1d). These results reveal that the TRPV1 channel is indeed expressed in the hippocampus and these channels are expressed in a functional state, in a small sub-population of neurons.

**TRPV1 expression in a sub-population of hippocampal neurons.** To investigate the identity of the potential sub-population of hippocampal neurons that express TRPV1, we immunostained dissociated hippocampal neurons and hippocampal sections with TRPV1 antibodies. In WT mouse hippocampal cultures immunostained with an N-terminal TRPV1 antibody, we found a small population of neurons with positive TRPV1 signal (Fig. 1e,f). These TRPV1-positive cells detected with the N-terminal antibody were absent in TRPV1 knockout cultures. When we used a C-terminal TRPV1 antibody, we identified the same small population of neurons detected by the N-terminal TRPV1 antibody. However, the C-terminal antibody still detected these neurons in TRPV1 knockout hippocampal cultures, although the intensity of signal was significantly reduced in TRPV1 knockout neurons (Fig. 1e,f). In combination with the qPCR data above, this suggests that the TRPV1 C-terminus persists in the TRPV1 knockouts. This sequence may correspond to a protein product generated by aberrant splicing in the TRPV1 knockouts, or to a remaining sequence of a naturally occurring splice isoform, such as VR.5′ sv. The VR.5′ sv splice isoform is more prevalent in the hippocampus than in DRG[21], and is also recognized by the C-terminal TRPV1 antibody, but not by the N-terminal TRPV1 antibody (Supplementary Fig. 3A). Because TRPV1 is homologous to TRPV2, TRPV3 and TRPV4, we also tested if the TRPV1 C-terminal antibody recognizes these additional TRPV isoforms. We found TRPV2 and TRPV4 signal in sub-populations of dissociated hippocampal neurons, but these neurons were distinct from those positive for TRPV1 recognized by the TRPV1 C-terminal antibody (Supplementary Fig. 3B,C). Interestingly, however, we found high co-localization of TRPV1 and TRPV3 in a subset of cells. The TRPV3 antibody did not recognize TRPV1 in over-expressing cells (Supplementary Fig. 3D), verifying that the TRPV3 antibody is isoform-specific and TRPV1 and TRPV3 are indeed co-expressed in a subset of hippocampal neurons. TRPV1 and TRPV3 knockouts have a similar reduction in Schaffer collateral LTP that is rescued by blocking GABAergic inhibition[22]. In addition TRPV1 and TRPV3 proteins interact, and their co-expression enhances TRPV1 responses to capsaicin[23]. Thus it is possible that these two isoforms cooperate in the same subset of inhibitory interneurons to affect LTP.

Immunohistochemistry of adult mouse hippocampal sections using the TRPV1 N-terminal and C-terminal antibodies yielded similar results. The N-terminal TRPV1 antibody identified a restricted sub-population of neurons expressing TRPV1, and such neurons were mainly present in the stratum oriens (Fig. 1g and Supplementary Fig. 3E). These TRPV1-expressing neurons were largely absent in TRPV1 knockouts (reduced by 80%). Similar to the observations made *in vitro*, the C-terminal TRPV1 antibody recognized the same small sub-population of TRPV1-expressing neurons in the stratum oriens of WT mice, but still recognized some cells in hippocampal sections of TRPV1 knockouts, where the number of cells per slice in TRPV1 knockouts was reduced by 40% compared to WT (Fig. 1h). The distribution of TRPV1-expressing neurons in the stratum oriens was reminiscent of somatostatin-expressing OLM (oriens-lacunosum-moleculare) neurons, which are known to be present in this region[24]. These neurons have large somata with axons extending to the stratum lacunosum moleculare[25,26], and horizontal dendrites projecting through the stratum oriens as well as to further layers of the CA1 region[27]. Indeed, we found that most TRPV1-expressing neurons in the stratum oriens also expressed somatostatin (Fig. 1i), suggesting that they are OLM cells.

**TRPV1 is expressed in oriens-lacunosum-moleculare neurons.** Using a lacZ reporter mouse, TRPV1 was previously reported to be expressed in the hippocampus in a very small population of reelin-expressing presumed Cajal–Retzius cells[10], which largely die before adulthood. However, reelin is also expressed in inhibitory interneurons in the hippocampus, and was recently found in a sub-population of somatostatin-positive OLM neurons[28]. We immunostained hippocampal sections with the N-terminal TRPV1 antibody, somatostatin and reelin and found that indeed, these markers co-localized in a sub-population of presumptive OLM cells in the stratum oriens (Fig. 2a,c). To further test the idea that TRPV1 is expressed in OLM neurons we immunostained for metabotropic glutamate receptor 7 (mGluR7), which is present in presynaptic terminals innervating OLM neurons[29,30]. We found that 81% of TRPV1-expressing neurons in the stratum oriens are densely decorated by detectable mGluR7 signal (Fig. 2b,c), indicating that these neurons correspond to OLM neurons in the hippocampus. Most TRPV1-positive neurons were also positive for the inhibitory interneuron marker GAD65, as expected (Fig. 2c), and for the C-terminal TRPV1 antibody, which recognizes the VR.5′ sv splice isoform that remains in the TRPV1 knockouts (Figs 1e,g,h and 2c). Only 20% of TRPV1-expressing cells were positive for parvalbumin in stratum oriens of PV-Tomato reporter mice, consistent with reports that a small subset of OLM neurons are parvalbumin-positive[28,31]. We did not detect any VIP-positive TRPV1-expressing neurons in sections from VIP-Tomato reporter mice (Fig. 2b,c).

**Figure 1 | Expression of TRPV1 in the hippocampus.** (**a**) qPCR showing relative mRNA expression levels of the TRPV1 pore region in DRG (D) and hippocampus (H), normalized to WT DRG levels. (**b**) qPCR showing relative mRNA expression levels of the TRPV1 N-T, C-T and pore regions (PR), and of the VR.5′sv splice isoform in DRG (left) and hippocampus (right) normalized to WT ($n = 3$ repetitions; error = s.e.m., significance determined by unpaired Student's $t$-test with Welch's correction; *$P < 0.05$, **$P < 0.01$, ***$P < 0.001$, ****$P < 0.0001$). (**c**) Western blot confirming that the N-terminal TRPV1 antibody recognizes TRPV1 in WT but not in TRPV1 KO homogenates from whole brain (upper) and hippocampus (lower). Tubulin served as a loading control; $n = 3$ repetitions. (**d**) Images of Fluo-4-based $Ca^{2+}$ responses in dissociated WT and TRPV1 knockout hippocampal cultures before, during and after field stimulation and after capsaicin addition (left); scale bar, 20 μm. Right panels show representative (coloured) and average (black) traces of Fluo-4 signal. Red arrows indicate A WT neuron that responded to 1 μM capsaicin treatment; no response was observed in TRPV1 KO cultures ($n = 18$ time-lapse recordings; 3 cultures) or to DMSO, used as a negative control ($n = 16$ recordings; three cultures). (**e**) The N-terminal TRPV1 antibody detects a subset of WT hippocampal neurons positive for TRPV1, and signal is absent in TRPV1 KO neurons ($n = 6$; two cultures). The C-terminal TRPV1 antibody detects the same sub-population of WT hippocampal neurons, but signal remains in TRPV1 KOs; scale bar, 20 μm. (**f**) Quantitation of N-terminal and C-terminal TRPV1 antibody signal in WT and TRPV1 KO hippocampal neurons ($n = 12$; 4 cultures; error = s.e.m., significance determined by unpaired Student's $t$-test with Welch's correction **$P < 0.01$, ***$P < 0.001$). (**g**) Immunostain of TRPV1 and DAPI showing TRPV1-positive cells in the stratum oriens in WT hippocampal sections, compared to TRPV1 KOs; scale bar, 20 μm. (**h**) Quantification of number of neurons positive for N-terminal or C-terminal TRPV1 antibody signal in WT and TRPV1 KO hippocampal sections ($n = 5$ sections each; error = s.e.m., significance determined by unpaired Student's $t$-test with Welch's correction *$P < 0.01$, **$P < 0.01$, ***$P < 0.001$). (**i**) Immunostain of TRPV1, somatostatin (SOM) and DAPI in WT and TRPV1 KO hippocampal sections; scale bar, 100 μm.

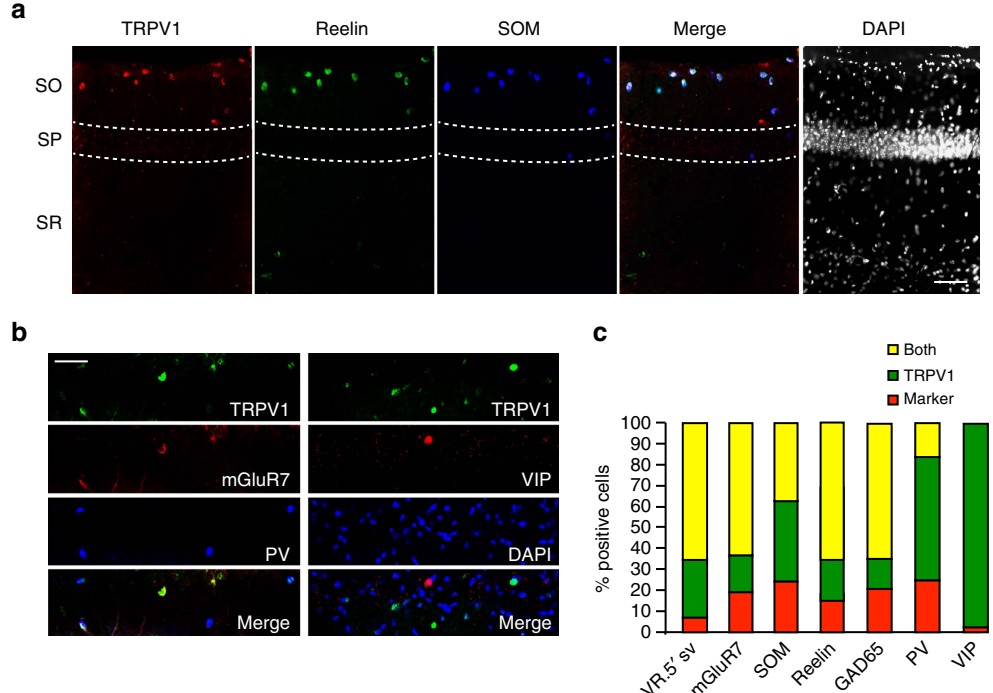

**Figure 2 | TRPV1 is expressed in OLM neurons in the hippocampus.** (**a**) Immunohistochemistry in hippocampal sections reveals a subset of neurons that express TRPV1, reelin and somatostatin in the stratum oriens; scale bar, 100 μm, SO: stratum oriens, SP: stratum pyramidale, SR: stratum radiatum). (**b**) Immunostains of the stratum oriens in hippocampal sections from PV-Tomato reporter mice for TRPV1 and mGluR7 (left) and from VIP-Tomato reporter mice with TRPV1 and DAPI (right); scale bar, 100 μm. (**c**) Quantitation of co-localization of immunofluorescence signal of different markers with TRPV1 neurons in the stratum oriens of hippocampal sections. VR.5′ sv quantitation indicates signal from the C-terminal TRPV1 antibody, compared to the N-terminal TRPV1 antibody (Fig. 1g). Number of sections analysed: TRPV1/VR.5′ sv $n = 5$, TRPV1/somatostatin $n = 3$, TRPV1/reelin $n = 3$, TRPV1/VIP $n = 3$, TRPV1/PV $n = 5$, TRPV1/GAD65 $n = 3$, TRPV1/mGluR7 $n = 3$.

We tested additional markers in hippocampal cultures, in which individual positive cells are more readily identified and the co-localization of markers can be easily quantified. We first tested if, in culture, TRPV1-expressing cells also co-express reelin and somatostatin; this was indeed the case, since ∼87% of TRPV1-expressing cells were also positive for reelin, and 62% were positive for somatostatin (Fig. 3a,c). Because reelin is an important signalling molecule necessary for the proper formation of cortical and hippocampal laminae during development[32], we tested if TRPV1 knockouts had any alterations in gross hippocampal (and cortical) architecture. In coronal brain sections from age-matched adult WT and TRPV1 knockout mice immunostained with the neuronal marker Neurotrace, and DAPI, to mark cell nuclei, we failed to detect any obvious structural alteration in the brain of TRPV1 knockout mice; cortical layers I–VI, DG and CA subdivisions of the hippocampus, and subcortical structures including the ventral posteromedial and posterolateral nuclei are present and well-defined (Supplementary Fig. 4).

We further confirmed that TRPV1-expressing neurons are inhibitory interneurons (and not Cajal–Retzius cells, as previously postulated[10]); 96% of TRPV1-expressing cells were positive for the inhibitory neuron marker GAD65. Only 14%, and none of TRPV1-expressing cells, respectively, were positive for parvalbumin, and VIP (Fig. 3a,c). 80% of TRPV1-expressing neurons *in vitro* had mGluR7 in terminals contacting them (Fig. 3b,c), further indicating that TRPV1 is specifically expressed in hippocampal OLM neurons[29,30]. The majority of TRPV1-positive neurons were also positive for the C-terminal TRPV1 antibody, which recognizes the VR.5′ sv splice isoform that remains in the TRPV1 knockouts (Figs 1e,f and 3c). There were

more non-TRPV1-expressing reelin, GAD65 and VIP neurons in dissociated cultures than in the stratum oriens region of hippocampal sections, most likely representing cells from other hippocampal regions.

In the peripheral nervous system, nerve growth factor (NGF) upregulates TRPV1 expression in nociceptors[33,34]. NGF is also retrogradely trafficked to septal cholinergic neurons—which innervate OLM cells[35] and promotes their survival[36]. We therefore also examined possible co-localization of TRPV1 with NGF. We found that TRPV1-expressing OLM neurons indeed express higher levels of NGF compared to surrounding neurons (Fig. 3d). Furthermore, treatment of hippocampal neurons with 1 ng ml$^{-1}$ NGF for 21–24 h increased TRPV1 expression, assayed both by immunostaining for TRPV1 (Fig. 3e) and by western blot of TRPV1, synaptobrevin-2 (as a negative control) and tubulin, used as a load control, in three biological replicates (Fig. 3f). This suggests that, as in the peripheral nervous system, NGF also upregulates TRPV1 in the hippocampus.

Interestingly, we found that TRPV1-expressing neurons are more highly innervated by excitatory vGluT1-positive synaptic boutons both *in vivo* (Fig. 3g) and *in vitro* (Fig. 3h), compared to surrounding neurons.

**TRPV1 promotes excitatory innervation of OLM neurons.** Although TRPV1 has not been reported as a synaptogenic factor, the high excitatory innervation of TRPV1-expressing OLM neurons in the hippocampus led us to hypothesize that TRPV1 itself may promote excitatory innervation. To test this, we first evaluated if, *in vitro*, weak and strong pharmacological activation of TRPV1 by 21–24 h incubation with either 50 nM or 1 μM

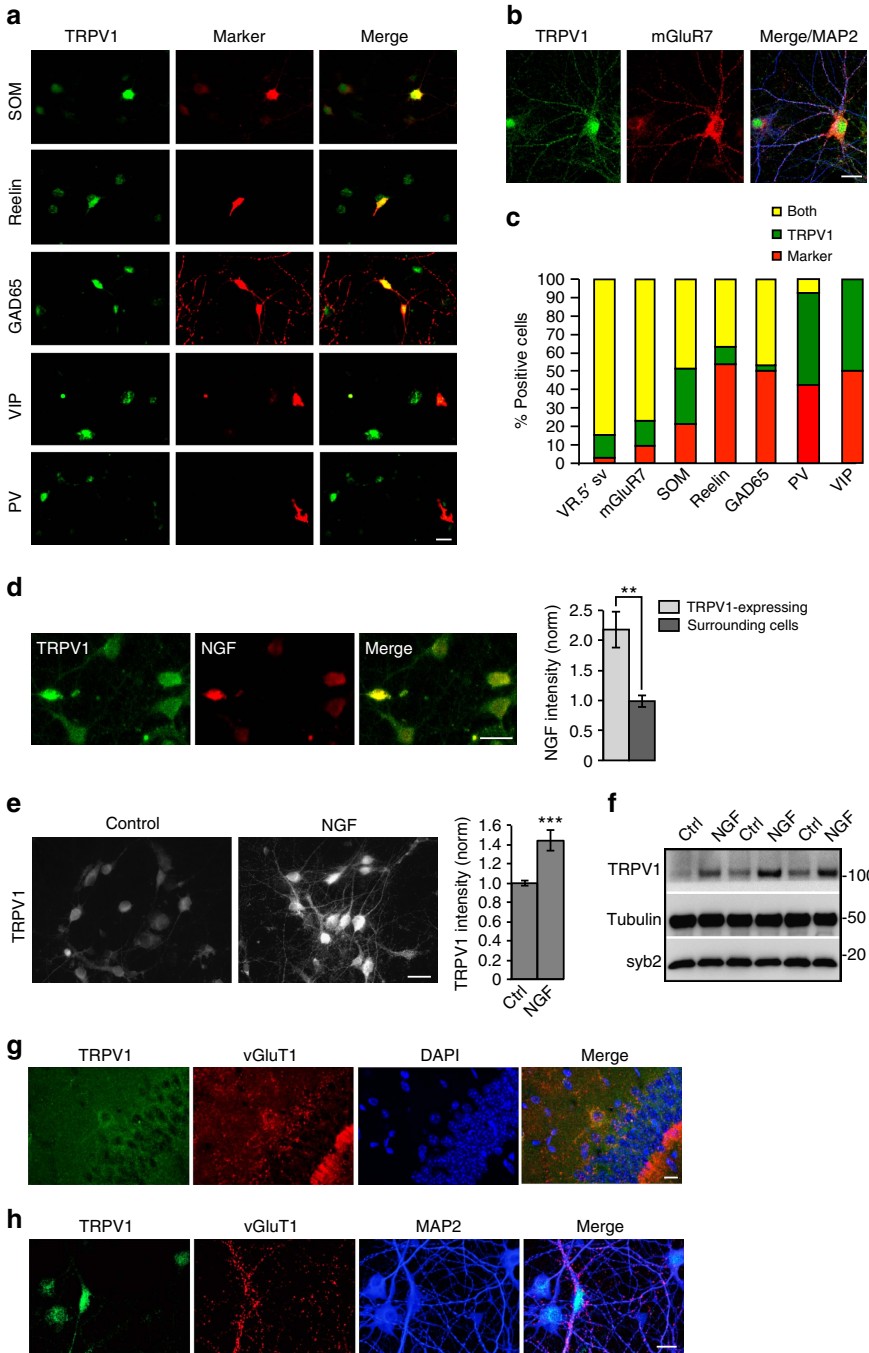

**Figure 3 | TRPV1-expressing hippocampal neurons are positive for OLM neuron markers and NGF upregulates TRPV1.** (**a**) Immunostains of hippocampal cultures for TRPV1 and Somatostatin (SOM), reelin, GAD65, VIP or Parvalbumin (PV). (**b**) Immunostain of hippocampal neuron culture for TRPV1, mGluR7 and MAP2; TRPV1-expressing neurons high mGluR7-positive innervation compared to surrounding neurons. (**c**) Quantitation of co-localization of immunofluorescence signal of different markers with TRPV1 neurons in hippocampal cultures. VR.5′ sv quantification indicates signal from the C-terminal TRPV1 antibody, compared to the N-terminal TRPV1 antibody (Fig. 1e). Cells/images analysed: TRPV1/VR.5′ sv $n = 54$, TRPV1/mGluR7 $n = 22$, TRPV1/somatostatin $n = 62$, TRPV1/reelin $n = 52$, TRPV1/GAD65 $n = 60$, TRPV1/PV $n = 26$, TRPV1/VIP $n = 22$, from 3 to 6 cultures; scale bar, 20 μm. (**d**) Immunostain (left) and quantitation (right) of hippocampal cultures showing that NGF is highly expressed in TRPV1-expressing hippocampal neurons, compared to surrounding neurons ($n = 15$ images from three cultures; error = s.e.m., significance determined by unpaired Student's $t$-test with Welch's correction, **$P < 0.01$). (**e**) Images of control hippocampal neuron cultures and cultures treated with 1 ng ml$^{-1}$ NGF at DIV12–13 for 21–24 h, immunostained for TRPV1 (left). Quantitation of TRPV1 immunofluorescence intensity normalized to control conditions is shown on the right ($n = 26$ (control), and 12 (NGF) from nine cultures; error = s.e.m., significance determined by unpaired Student's $t$-test with Welch's correction, ***$P < 0.001$). (**f**) Western blot of TRPV1 in three repetitions of treatment of cultured hippocampal neurons with NGF compared to control. The levels of synaptobrevin-2 (used as a negative control) are unchanged. Tubulin serves as a load control. (**g**) Immunostain of a hippocampal section with TRPV1, vGluT1 and DAPI, showing high vGluT1 innervation of a TRPV1-expressing neuron in the stratum oriens; scale bar = 20 μm. (**h**) Immunostain of a dissociated hippocampal neuron culture with TRPV1, vGluT1 and MAP2 (to mark neuronal cell bodies and processes), showing high vGluT1 innervation of a TRPV1-expressing neuron; scale bar, 20 μm.

capsaicin, or with the TRPV1 antagonist SB-366791, would have any influence on innervation of TRPV1-expressing cells. Using immunocytochemistry we quantified the number of vGluT1- and vGAT-positive excitatory and inhibitory terminals, respectively, contacting TRPV1-expressing neurons compared to surrounding neurons. We also quantified the fluorescence intensity of both markers, and the concomitant signal from an antibody against the lumenal domain of synaptotagmin1, which was taken up by recycling synaptic vesicles during depolarization and live

labelling, before fixation (syt1u). The intensity of syt1u signal at vGluT1 or vGAT-positive terminals acts as a readout of excitatory and inhibitory presynaptic strength.

On average, TRPV1-expressing neurons had more than twice the number of excitatory contacts compared to surrounding neurons (Fig. 4a–c and Supplementary Fig. 5). The number of excitatory contacts increased after 21–24 h 50 nM capsaicin treatment, and increased further with a higher concentration of 1 μM capsaicin (Fig. 4c). Interestingly, 1 μM capsaicin also

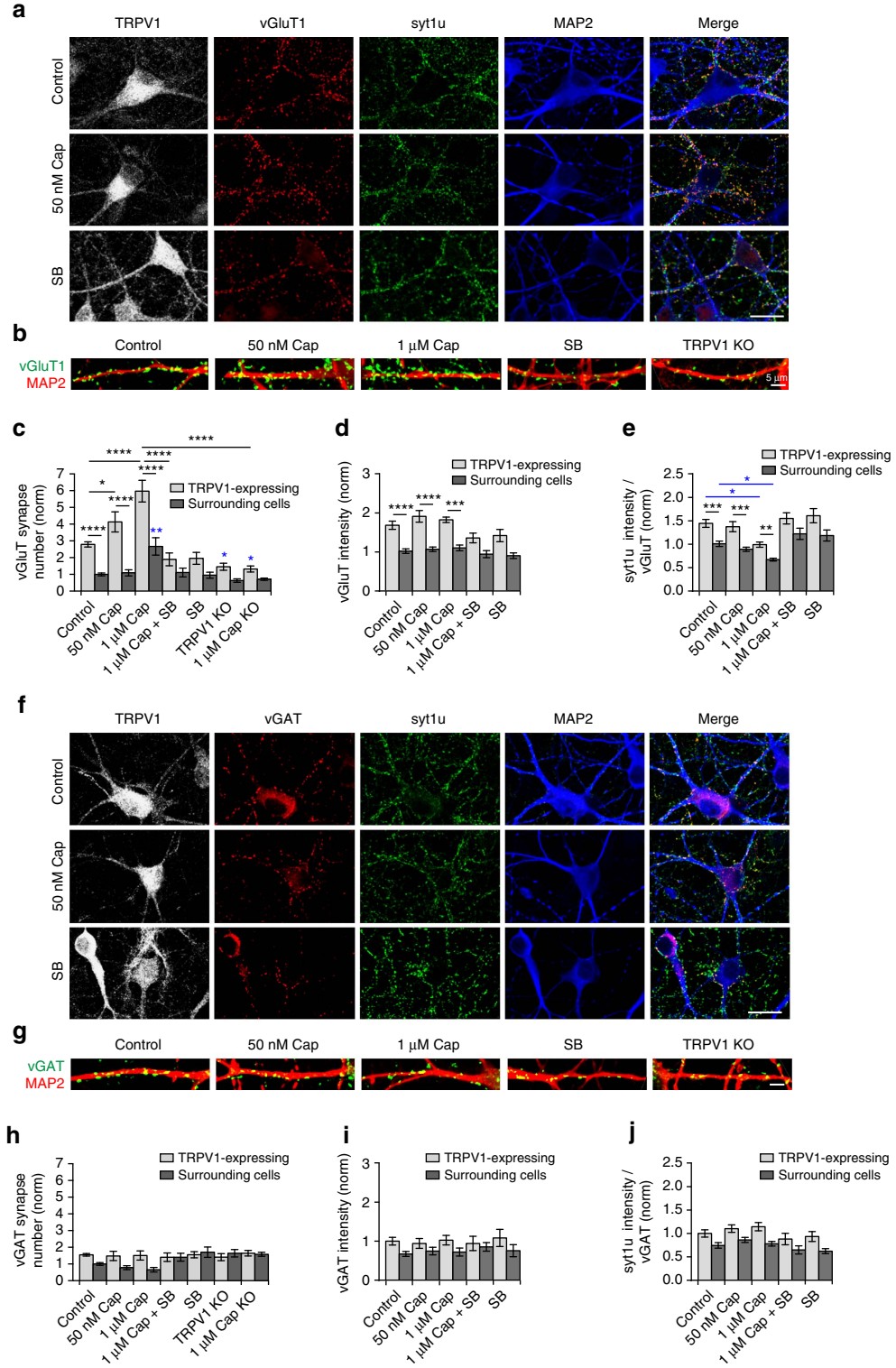

doubled the number of vGluT1 puncta on surrounding neurons, and this effect was absent in TRPV1 knockouts; larger field images including both TRPV1-expressing and non-expressing neurons following capsaicin treatment are shown in Supplementary Fig. 5A. Treatment with 50 nM or 1 µM capsaicin for only 4 h did not further increase excitatory innervation in TRPV1-expressing on non-expressing neurons (Supplementary Fig. 5B,C). VGluT1 puncta intensity and syt1 uptake signal was also higher in TRPV1-expressing neurons compared to surrounding neurons, but did not change significantly following capsaicin treatment (Fig. 4d,e), except for syt1 fluorescence intensity following 1 µM capsaicin treatment, which decreased slightly in both TRPV1-expressing and non-expressing neurons (Fig. 4e). Treatment of cultures with a selective TRPV1 antagonist, SB-366791 (1 µM) decreased excitatory synapse number only on TRPV1-expressing neurons, and reverted the effect of capsaicin treatment (Fig. 4a–c). In addition, the same sub-population of OLM neurons in TRPV1 knockout cultures (identified by C-terminal TRPV1 antibody signal, which remains in the knockouts) had decreased excitatory innervation compared to WT TRPV1-expressing neurons, and capsaicin had no effect on excitatory synapse number on TRPV1 knockout neurons (Fig. 4b,c).

Inhibitory innervation of TRPV1-expressing hippocampal neurons was similar to the surrounding neurons (Fig. 4f–h; note that the scale of graphs in Fig. 4c–e,h–j is the same for comparison) and was unchanged by 1 µM capsaicin treatment, blockade of TRPV1 channels with SB-366791, or knockout of TRPV1 (Fig. 4h). The intensity of vGAT and syt1 uptake signal in inhibitory terminals contacting TRPV1-expressing cells was also similar to that of surrounding neurons, and unchanged by capsaicin, SB-366791 treatment or knockout of TRPV1 (Fig. 4i,j).

We further used the TRPV1 knockout model to examine the effects of TRPV1 on excitatory and inhibitory synapse number in OLM interneurons identified by co-expression of somatostatin and reelin. WT OLM neurons co-expressing somatostatin and reelin had more than twice as many vGluT1 synapses as surrounding neurons (Fig. 5a,b), and the number of synapses increased two-fold following 1 µM capsaicin treatment. The number of excitatory terminals on TRPV1 knockout OLM neurons, by contrast, was reduced compared to WT; there was no significant difference in the number of excitatory synapses on OLM neurons versus surrounding neurons in TRPV1 knockout cultures. In addition, the capsaicin-induced increase in excitatory innervation of OLM neurons in WT cultures, was absent in TRPV1 knockout cultures (Fig. 5b; larger field images of WT and TRPV1 knockout neurons including TRPV1-expressing and non-expressing surrounding neurons following capsaicin treatment are shown in Supplementary Fig. 5D). By contrast, knockout of TRPV1 did not significantly affect the number of inhibitory synapses on OLM interneurons, and 1 µM capsaicin also had no effect on inhibitory synapse number on OLM neurons in WT (or knockout) cultures (Fig. 5c,d). Thus, TRPV1 appears to specifically promote excitatory, and not inhibitory, innervation of OLM neurons.

To verify this finding electrophysiologically, we recorded from putative OLM neurons—identified as neurons with large somata located in the stratum oriens—in hippocampal slices from WT and TRPV1 knockout mice. Because OLM neurons represent the vast majority of inhibitory interneurons in the stratum oriens of hippocampal slices[37,38], we identified cells in this region by morphology for whole-cell patch clamp experiments. We found that mEPSC frequency was significantly reduced in putative OLM neurons in TRPV1 knockout hippocampal slices compared to those in WT slices (Fig. 6a,b), while mEPSC amplitude was unchanged (Fig. 6c). To confirm that these neurons expressed TRPV1, we then recorded currents following perfusion of 1 µM capsaicin. In WT slices, 9 out of 12 recorded neurons responded to capsaicin with inward currents ranging from 100 to 400 pA (Fig. 6d). In TRPV1 knockout slices out of 19 recorded neurons, we observed no significant inward currents in response to capsaicin perfusion, except for one cell with a 400 pA depolarizing response, and another with a 200 pA hyperpolarizing response, which were identified as outliers by ROUT analysis of current peak amplitude. These data confirm our findings that TRPV1 promotes excitatory innervation of OLM neurons.

**Heterologous TRPV1 expression increases excitatory inputs.** We next sought to determine if TRPV1 can promote excitatory synapse formation in cells that do not normally express it. To test this, we transfected rat hippocampal neurons at DIV2 with TRPV1-IRES-GFP (or EGFP as a control) using Lipofectamine 2000, and assayed synapse number and presynaptic strength on transfected neurons at DIV12–13. The transfection efficiency achieved using this technique is relatively low, with ∼1–5% of cells transfected. Thus virtually all presynaptic terminals contacting transfected neurons are 'wild-type', where TRPV1 is only overexpressed post-synaptically. Transfection of TRPV1-IRES-GFP increased the number of excitatory terminals contacting transfected neurons compared to EGFP controls (Fig. 7a,b) (but to a lesser extent than TRPV1-expressing OLM neurons compared to surrounding neurons). In addition the fluorescence intensity of vGluT1 in presynaptic terminals (Fig. 7c)

**Figure 4 | Post-synaptic TRPV1 promotes excitatory innervation. (a)** Quadruple immunocytochemistry images of hippocampal neurons immunostained for TRPV1, vGluT1, syt1 lumenal antibody internalized by recycling synaptic vesicles during depolarization, and MAP2, in the indicated conditions; scale bar, 20 µm. **(b)** Higher magnification images of vGluT1 and MAP2 on TRPV1-expressing neurons in hippocampal cultures; scale bar, 5 µm. **(c)** Quantification of excitatory synapse number on TRPV1-positive hippocampal neurons normalized to surrounding cells in the indicated conditions. Black asterisks indicate significance between TRPV1-expressing and surrounding neurons in each condition. Blue asterisks indicate significance relative to control for either TRPV1-expressing cells or surrounding cells. **(d)** Quantitation of vGluT1, and syt1 uptake **(e)** intensity in presynaptic terminals contacting TRPV1-expressing neurons, normalized to surrounding neurons, in the indicated conditions. Images used for quantitation were: WT control $n = 24$, WT 50 nM cap. $n = 12$, WT 1 µM cap. $n = 9$, WT 1 µM cap. + 1 µM SB $n = 12$, WT 1 µM SB $n = 12$, TRPV1 KO control $n = 9$, TRPV1 KO 1 µM cap. $n = 9$; from five different cultures. **(f)** Immunostains of TRPV1, vGAT, lumenal syt1 antibody internalized by recycling synaptic vesicles during depolarization, and MAP2 in the indicated conditions; scale bars, 20 µm. **(g)** Higher magnification images of vGAT and MAP2 immunostaining; scale bars, 5 µm. **(h)** Quantitation of inhibitory synapse number (number of vGAT-positive puncta) on TRPV1-expressing neurons normalized to surrounding cells. **(i)** Quantitation of vGAT intensity in presynaptic terminals contacting TRPV1-expressing neurons, normalized to surrounding neurons, in the indicated conditions. **(j)** Quantitation of syt1 uptake in depolarizing conditions in vGAT-positive terminals contacting TRPV1-expressing neurons, normalized to surrounding neurons, in the indicated conditions. Images used for quantitation were: WT control $n = 23$, WT 50 nM cap. $n = 11$, WT 1 µM cap. $n = 9$, WT 1 µM cap. + 1 µM SB $n = 12$, WT 1 µM SB $n = 12$, TRPV1 KO control $n = 9$, TRPV1 KO 1 µM cap. $n = 9$; from 5 cultures. Error = s.e.m., significance determined by one-way ANOVA with Tukey's *post hoc* test for multiple comparisons, *$P < 0.05$, **$P < 0.01$, ***$P < 0.001$, ****$P < 0.0001$; fluorescence intensity and synapse number for all conditions was normalized to the corresponding average value for surrounding neurons in WT cell cultures.

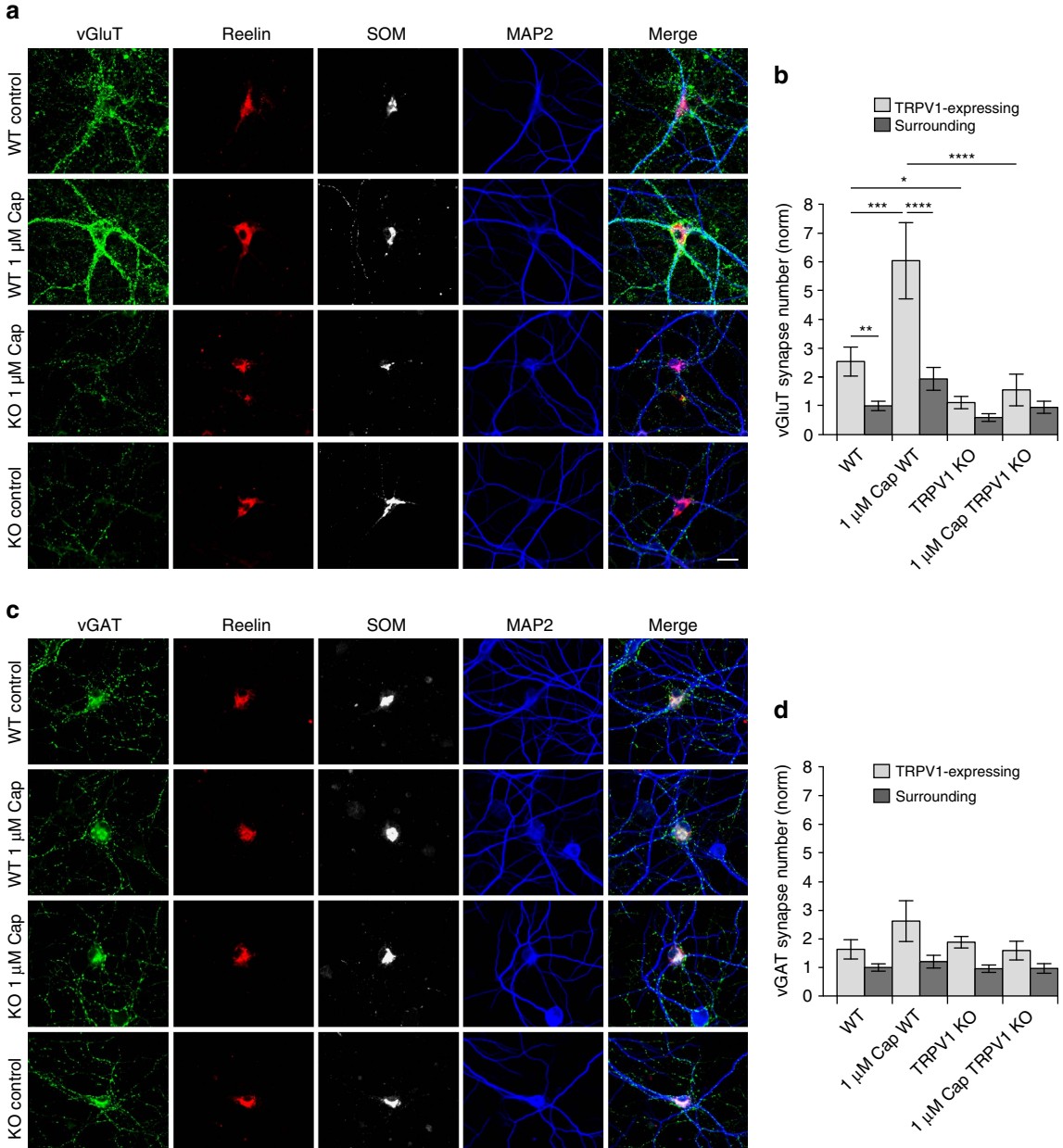

**Figure 5 | TRPV1 promotes excitatory innervation of OLM neurons.** (**a**) Quadruple immunostains of WT and TRPV1 KO hippocampal neurons with vGluT1 to assay excitatory presynaptic innervation, somatostatin and reelin to identify OLM cells, and MAP2 to identify neuronal somas and dendritic processes, in the indicated conditions. Merge indicates combination of vGluT, reelin and MAP-2 channels; scale bar, 20 μm. (**b**) Quantitation of excitatory synapse (vGluT1 puncta) number, in the indicated conditions, normalized to surrounding non-TRPV1-expressing cells. TRPV1 KO OLM neurons have reduced excitatory innervation, compared to WT ($n = 20$ images for all conditions from three cultures). (**c**) Quadruple immunostain of WT and TRPV1 KO hippocampal neurons for vGAT, to mark inhibitory synapses, reelin, somatostatin, and MAP2. Merge indicates combination of vGAT, reelin and MAP2 channels; scale bar, 20 μm. (**d**) Quantitation of inhibitory synapse (vGAT puncta) number, in the indicated conditions, normalized to surrounding non-TRPV1-expressing cells ($n = 20$ images for all conditions from 2 cultures). Error = s.e.m., statistical significance determined by one-way ANOVA with Tukey's *post hoc* test for multiple comparisons, *$P < 0.05$, **$P < 0.01$, ***$P < 0.001$, ****$P < 0.0001$.

and the co-localizing syt1 uptake immunofluorescence signal (Fig. 7d) were also increased on TRPV1-expressing neurons compared to controls. Blockade of the TRPV1 channel with 1 μM SB-366791 on DIV4 reverted these effects to control levels. We also measured mEPSC frequency and amplitude electrophysiologically, and found that mEPSC frequency (which is thought to reflect synapse number or release probability), but not mEPSC amplitude, was higher in TRPV1-transfected neurons than in EGFP-transfected control neurons (Fig. 7e).

We also tested inhibitory synapse formation and strength on neurons over-expressing TRPV1 in cultured hippocampal neurons. Although we found no change in inhibitory synapse number on TRPV1-IRES-GFP-expressing neurons (Fig. 7f,g), vGAT fluorescence intensity on these neurons was slightly decreased compared to EGFP controls, and was unchanged by blockade of the TRPV1 channel (Fig. 7h). Syt1 antibody uptake signal at vGAT synapses was also significantly reduced compared to control, and this reduction was absent in the

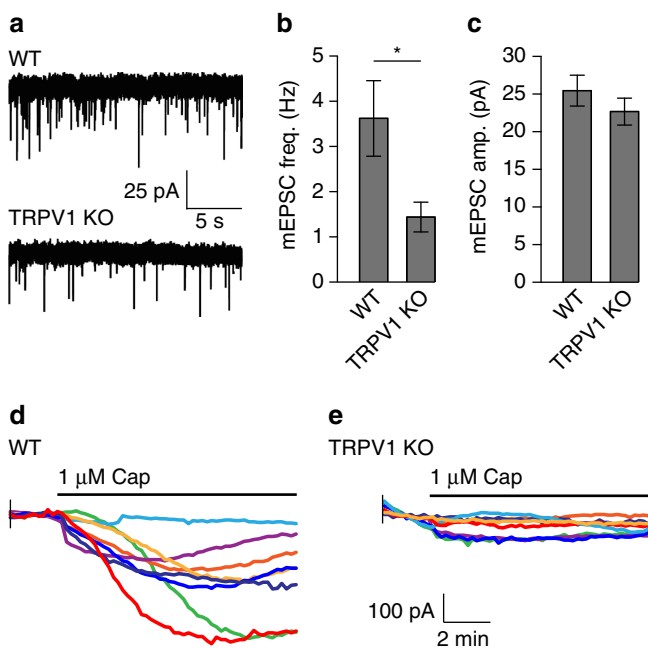

**Figure 6 | TRPV1 knockouts have reduced mEPSC frequency in OLM neurons.** (**a**) Representative mEPSC traces from recordings of putative OLM neurons in the stratum oriens of hippocampal slices from WT and TRPV1 knockouts. (**b**) Quantitation of mEPSC frequency, and amplitude (**c**) in WT and TRPV1 knockout OLM neurons (WT $n = 19$ neurons, TRPV1 KO $n = 16$; error = s.e.m., significance determined by unpaired Student's t-test with Welch's correction, *$P < 0.05$). (**d**) Current responses to perfusion of $1\,\mu$M capsaicin in recorded putative OLM neurons in WT and TRPV1 knockout (**e**) hippocampal slices (WT $n = 12$ neurons, TRPV1 KO $n = 19$).

presence of $1\,\mu$M SB-366791 added at DIV4 to block TRPV1 channels (Fig. 7i). The frequency and amplitude of mIPSCs recorded from TRPV1-transfected neurons were unchanged compared to EGFP-transfected control neurons, corroborating the unchanged number of inhibitory terminals innervating TRPV1-transfected neurons compared to EGFP-transfected control neurons (Fig. 7j). These results demonstrate that TRPV1—expressed in cells in which it is normally absent—promotes excitatory, but not inhibitory, innervation of these cells.

**TRPV1-induced innervation requires Ca²⁺, NGF and BDNF.** What are the molecular and cellular mechanisms by which TRPV1 promotes excitatory synapse formation? NGF increases TRPV1 expression in hippocampal neurons (Fig. 3e,f), and the high levels of NGF in OLM neurons may therefore promote TRPV1 expression specifically in these cells. Activated TRPV1 increases calcium entry in the cells in which it is expressed, which may lead to increased activity of these cells, increased coincident pre- and post-synaptic firing and subsequent strengthening of existing connections via a Hebbian mechanism[39]. Increased activity in TRPV1-expressing neurons may then activate other activity-dependent signalling molecules that promote excitatory synapse formation. Brain-derived neurotrophic factor (BDNF), for example, has been reported to be upregulated by TRPV1 in DRG neurons in response to inflammation[40], and activation of TRPV1 increases BDNF release[41]. BDNF stimulates synaptogenesis in hippocampal neurons[42], and increases excitatory, but not inhibitory, presynaptic strength[43] and therefore may be important for TRPV1-mediated excitatory innervation. Indeed, we found that TRPV1-expressing neurons had increased BDNF levels compared

to non-TRPV1-expressing neurons (Fig. 8a,b). In addition, WT OLM neurons had a greater number and intensity of BDNF puncta than TRPV1 knockouts (Fig. 8c,d).

To test if NGF, BDNF or Ca²⁺ entry through TRPV1 channels are necessary for the increase in excitatory synaptic innervation mediated by TRPV1, we blocked Ca²⁺ entry (by addition of 2 mM EGTA), NGF (with bath application of $1\,\mu$g ml⁻¹ TrkA-Fc), or BDNF (with bath application of $0.4\,\mu$g ml⁻¹ TrkB-Fc) during incubation with $1\,\mu$M capsaicin for 21–24 h, which when applied alone normally induces a significant increase in excitatory innervation, specifically in TRPV1-expressing neurons. Interestingly, we found that all of these treatments blocked the TRPV1-mediated synaptogenic effect (Fig. 8e,f). Capsaicin plus TrkA-Fc, or capsaicin plus TrkB-Fc, both reduced capsaicin-induced TRPV1-mediated excitatory innervation to control levels. Addition of 2 mM EGTA significantly reduced excitatory innervation below control levels, both in the presence and absence of $1\,\mu$M capsaicin, suggesting that calcium influx is necessary for the maintenance of synapses in general, and not just for synapse formation induced by TRPV1. Together, these results suggest a pathway whereby NGF upregulates TRPV1 in OLM neurons, which in turn increases calcium influx via activated TRPV1. This activates BDNF, which promotes excitatory innervation of these cells. In support of this pathway, we found that overexpression of TRPV1 does not upregulate NGF (Fig. 8g) but does upregulate BDNF (Fig. 8h), in transfected neurons.

We further tested if increasing neuronal activity alone is sufficient to induce excitatory synapse formation on OLM neurons. To increase neuronal activity, we transduced hippocampal cultures with ChR2-EYFP AAV, or EGFP AAV as a control, and then delivered pulses of blue light via light-emitting diode (LED) in the incubator for 24 h. Although LED stimulation was sufficient to depolarize ChR2-EYFP expressing neurons (Supplementary Fig. 6A), we did not observe an increase in excitatory innervation of TRPV1-expressing hippocampal neurons that were transduced with ChR2-EYFP, compared to EGFP controls (Supplementary Fig. 6B,C). However, in these experiments the majority of cells were transduced and the entire culture was stimulated at low light intensity. It is therefore still possible that specific differential levels of activity between cells may promote excitatory innervation in a Hebbian manner.

**TRPV1-induced OLM neuron innervation is necessary for LTP.** The deficient excitatory innervation of OLM neurons in TRPV1 knockouts that we discovered may explain why TRPV1 knockouts have decreased Schaffer collateral hippocampal LTP compared to WT mice[11]. OLM cells directly inhibit the temporoammonic pathway at distal dendrites of CA1 pyramidal cells, and disinhibit the Schaffer collateral pathway via contact with other intermediate inhibitory interneurons (Fig. 9a)[35]. Less excitatory innervation of OLM cells would therefore lead to decreased disinhibition (ie, more inhibition of CA1 pyramidal neurons) and reduced LTP. To test this idea, we first compared LTP in the Schaffer collateral pathway of acute hippocampal slices from TRPV1 knockout and WT mice (Fig. 9b). We found a significant decrease in LTP in the Schaffer collateral pathway in recordings from TRPV1 knockouts compared to WT (Fig. 9c), suggesting that TRPV1 plays a role in normal hippocampal functioning and plasticity, as previously reported[11,14,15,22].

Nicotine is known to promote the induction of LTP in the Schaffer collateral pathway[44,45]. Recently it was reported that in the hippocampus, OLM neurons exclusively express the α2 nicotinic receptor subunit, and nicotine-dependent LTP induction in the Schaffer collateral pathway is mediated

specifically through an OLM-dependent disinhibitory mechanism[35]. We reasoned that if TRPV1 knockouts have reduced LTP specifically due to reduced glutamatergic innervation of OLM neurons, the LTP deficit in TRPV1 knockouts should be rescued by activating OLM neurons with nicotine, bypassing the TRPV1-mediated excitatory innervation

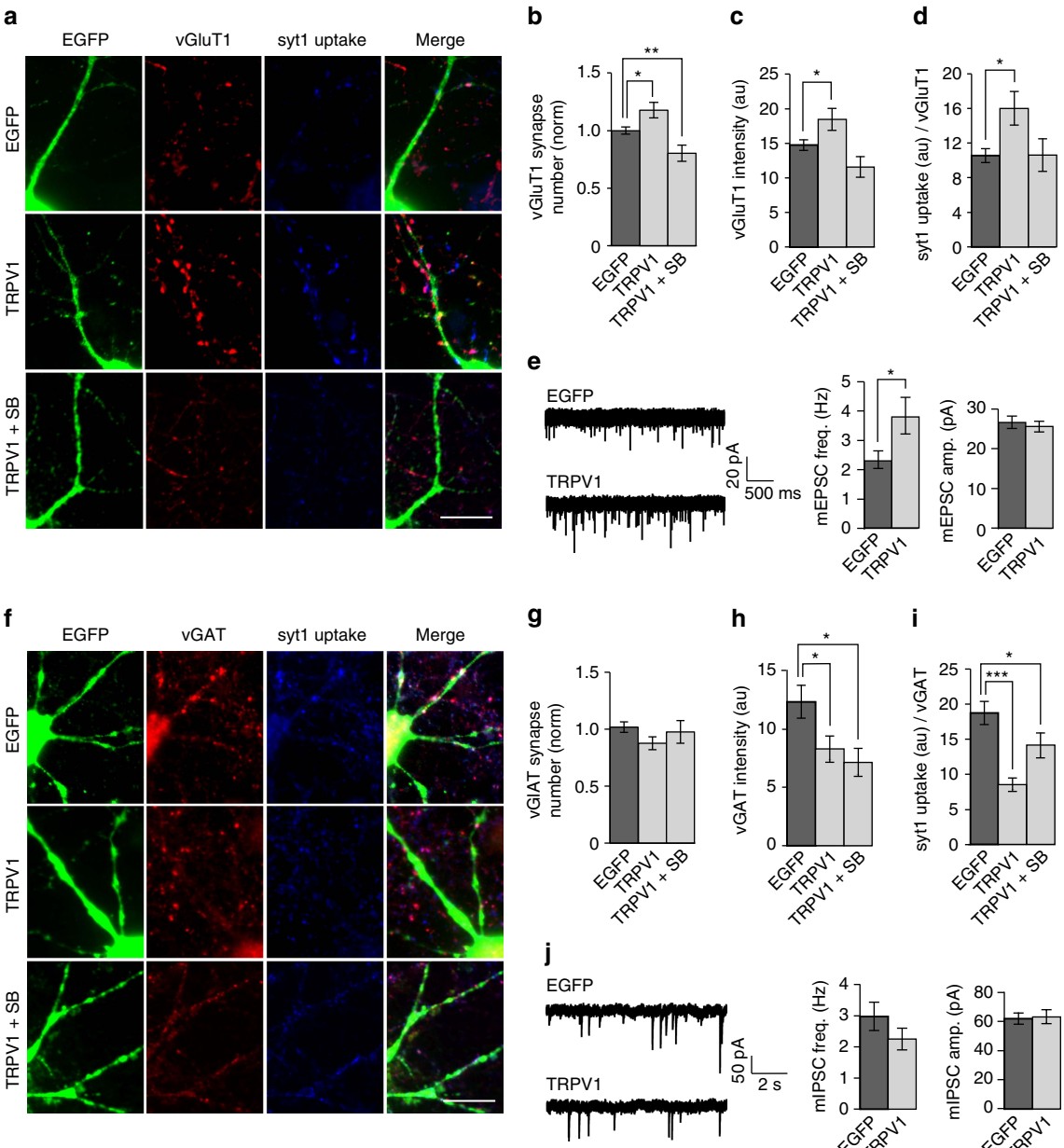

**Figure 7 | TRPV1 overexpression in neurons increases excitatory innervation.** (**a**) Immunostains of TRPV1-IRES-GFP and EGFP transfected neurons in the indicated conditions with vGluT1 and lumenal syt1 antibody internalized by recycling synaptic vesicles during depolarization; scale bars, 10 μm. (**b**) Quantitation of excitatory synapse number (number of vGluT1 puncta) on TRPV1-expressing cells with and without treatment with 1 μM SB-366791 to block TRPV1 channels, compared to EGFP controls. (**c**) Quantitation of vGluT1 signal intensity (corresponding to number of vGluT1 vesicles per terminal) in the indicated conditions. (**d**) Quantitation of lumenal syt1 antibody uptake in depolarizing conditions (corresponding to the total number of recycling synaptic vesicles in excitatory terminals) in vGluT1-positive terminals (neurons/images quantified were: EGFP $n = 60$, TRPV1-IRES-GFP $n = 30$, TRPV1-IRES-GFP + 1 μM SB-366791 DIV4 $n = 15$; from 5 to 20 different neuronal cultures). (**e**) TRPV1-IRES-GFP-expressing neurons have increased mEPSC frequency compared to control EGFP-expressing neurons, and unchanged amplitude, consistent with an increase in synapse number in TRPV1-over-expressing neurons (cells used for analysis: EGFP $n = 29$, TRPV1-IRES-GFP $n = 27$). (**f**) Immunostain of TRPV1-IRES-GFP or EGFP transfected neurons in the indicated conditions with vGAT and syt1 internalized by recycling synaptic vesicles during depolarization; scale bars, 10 μm. (**g**) Quantitation of inhibitory synapse number (number of vGAT puncta) on TRPV1-expressing cells with and without treatment with 1 μM SB-366791 to block TRPV1 channels, compared to EGFP. (**h**) Quantitation of vGAT signal intensity (corresponding to number of vGAT vesicles per terminal). (**i**) Quantitation of lumenal syt1 antibody uptake in depolarizing conditions (corresponding to the total number of recycling synaptic vesicles in inhibitory terminals) in vGAT-positive terminals (neurons/images quantified were: EGFP $n = 60$, TRPV1-IRES-GFP $n = 27$, TRPV1-IRES-GFP + 1 μM SB-366791 DIV4 $n = 15$; from five cultures). (**j**) TRPV1-IRES-GFP-expressing neurons have no change in mIPSC frequency or amplitude, compared to control EGFP-expressing neurons (cells used for analysis: EGFP $n = 27$, TRPV1-IRES-GFP $n = 24$). Error = s.e.m.; significance determined by unpaired Student's $t$-test *$P < 0.05$, **$P < 0.01$, ***$P < 0.001$.

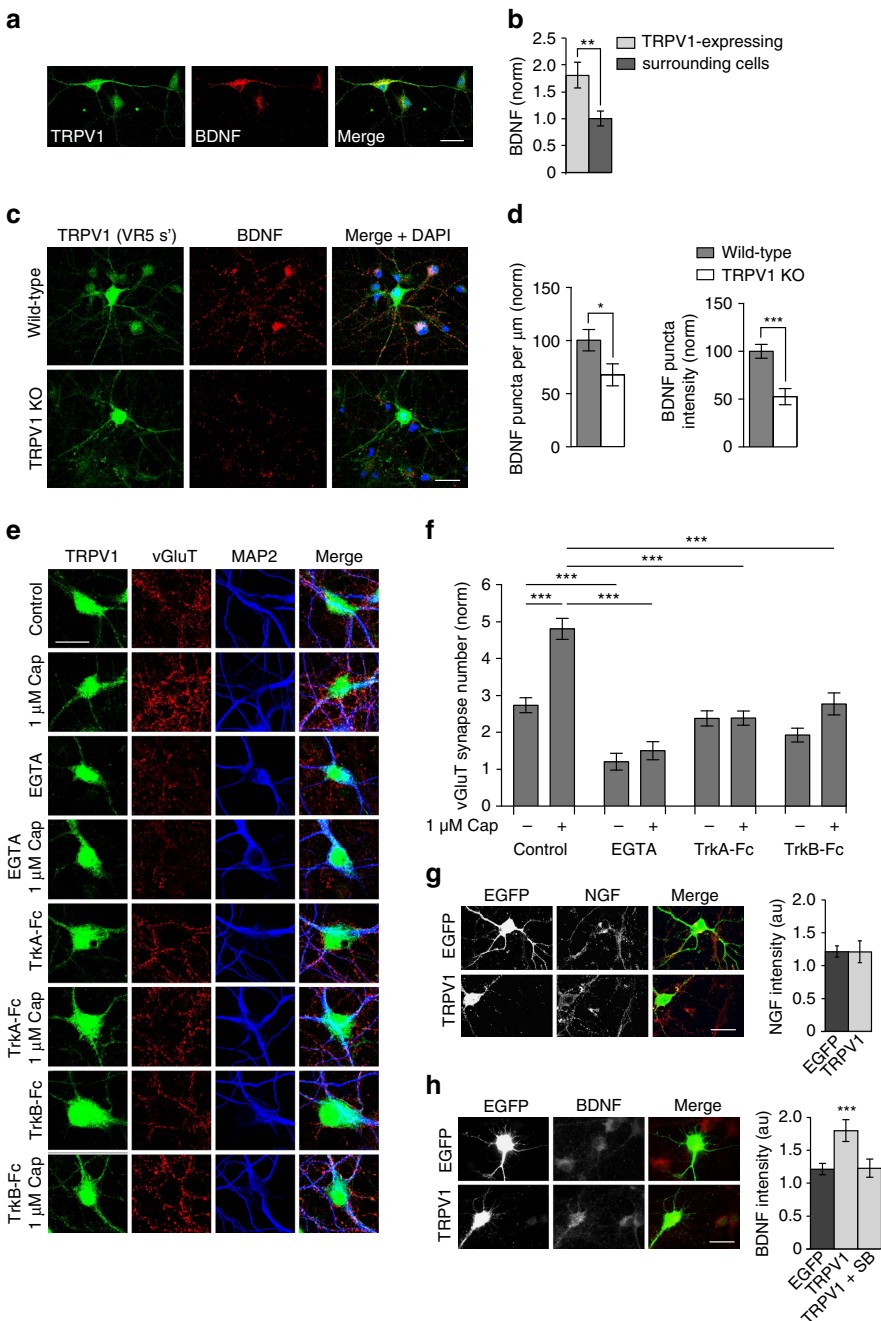

**Figure 8 | TRPV1-induced excitatory synaptogenesis requires calcium influx, NGF and BDNF. (a)** Immunostain of rat hippocampal cultures with TRPV1 and BDNF. **(b)** Quantitation of BDNF signal in somas of TRPV1-expressing neurons normalized to surrounding non-TRPV1-expressing cells ($n = 15$ images from three cultures; error = s.e.m., significance determined by unpaired Student's $t$-test with Welch's correction, $**P < 0.01$). **(c)** Immunostain of WT and TRPV1 knockout mouse cultures with the C-terminal TRPV1 antibody (which detects a remainins splice isoform in the TRPV1 knockouts and marks TRPV1-expressing cells—Fig. 1e,f), BDNF, and DAPI. **(d)** Quantitation of BDNF puncta/μm (left panel) and BDNF puncta intensity (right panel) on OLM neurons marked by the C-terminal TRPV1 antibody, in WT and TRPV1 knockout cultures, normalized to WT. **(e)** Immunostains of TRPV1, vGluT and MAP2 in cultures in control conditions, and in cultures treated with capsaicin in the presence of absence of 2 mM EGTA to block calcium influx, 1 μg ml$^{-1}$ TrkA-Fc to scavenge NGF, or 0.4 μg ml$^{-1}$ TrkB-Fc to scavenge BDNF. **(f)** Quantitation of excitatory synapse number (vGluT puncta number) on TRPV1-expressing hippocampal neurons in the indicated conditions. Images used for quantitation were: control $n = 28$, 1 μM cap. $n = 14$, 2 mM EGTA $n = 21$, 2 mM EGTA + 1μM cap. $n = 21$, 1 μg ml$^{-1}$ TrkA-Fc $n = 21$, 1 μg ml$^{-1}$ TrkA-Fc + 1 μM cap. $n = 21$, 0.4 μg ml$^{-1}$ TrkB-Fc $n = 19$, 0.4 μg ml$^{-1}$ TrkB-Fc + 1 μM cap. $n = 20$; from four cultures. Error = s.e.m., significance determined by one way ANOVA with Tukey's *post hoc* test for multiple comparisons, $***P < 0.001$. **(g)** Immunostains of EGFP and TRPV1-IRES-GFP transfected hippocampal neurons with NGF. Quantitation of NGF intensity in somas of transfected neurons is indicated on the right ($n = 12$ images from three cultures; error = s.e.m., significance determined by unpaired Student's $t$-test with Welch's correction). **(h)** Immunostains of EGFP and TRPV1-IRES-GFP transfected hippocampal neurons with BDNF. Quantification of BDNF intensity in somas of EGFP and TRPV1-IRES-GFP transfected neurons with and without addition of 1 μM SB-366791 2 days after transfection, is indicated on the right ($n = 13$ images from three cultures; error = s.e.m., significance determined by unpaired Student's $t$-test with Welch's correction, $***P < 0.001$). Scale bars, 20 μm in all panels.

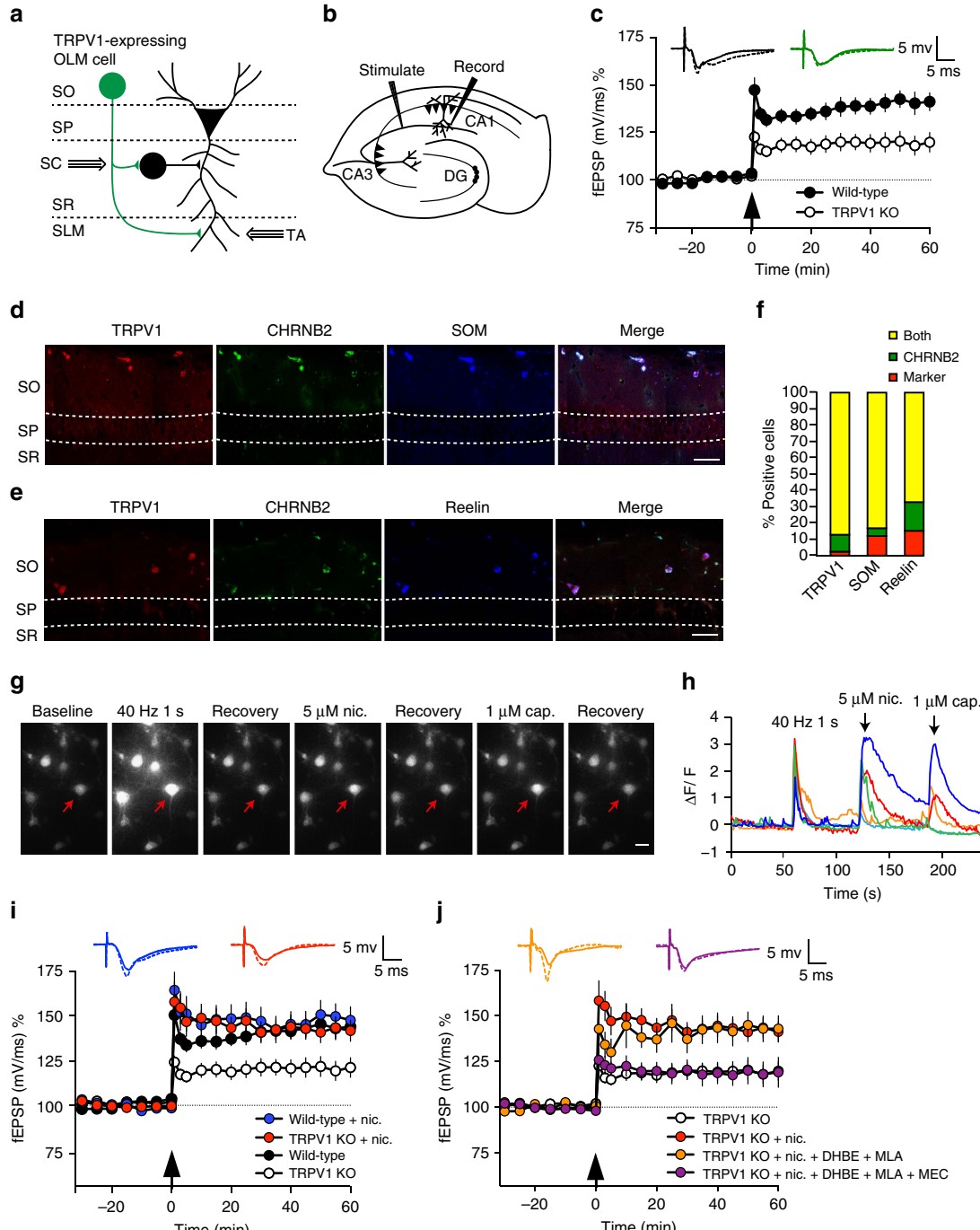

**Figure 9 | TRPV1 affects synaptic plasticity in the Schaffer collateral pathway via promoting innervation of OLM cells.** (**a**) Scheme of putative connections of a TRPV1-expressing OLM cell in the stratum oriens. OLM cells inhibit the temporoammonic (TA) pathway from the entorhinal cortex, and disinhibit the Schaffer collateral pathway from the CA3 region, thus promoting CA3 input, and reducing EC input, to CA1 pyramidal cells. (**b**) Illustration of stimulating and recording electrode positions for LTP recordings performed in the Schaffer collateral pathway. (**c**) LTP induced by 1XTET-LTP in the Schaffer collateral pathway is reduced in TRPV1 KOs compared WT ($n = 10$ sections each from WT and TRPV1 KO mice; error = s.e.m.). (**d**) Immunostain of hippocampal sections from a β2-EGFP nAChR mouse with TRPV1 and SOM, and with TRPV1 and reelin (**e**); TRPV1 and β2-EGFP nAChRs are expressed in the same population of OLM neurons. (**f**) Quantitation of β2-EGFP nAChR with TRPV1, SOM, and reelin-expressing cells in the stratum oriens ($n = 8$, 4 and 3 images for TRPV1, SOM, and reelin, respectively, from four hippocampal sections). (**g**) Calcium imaging of Fluo-4 signal in dissociated hippocampal neurons stimulated sequentially with 5 μM nicotine and 1 μM capsaicin. 80% of cells that responded to capsaicin also responded to nicotine; of 112 cells that responded to field stimulation, 17 responded to nicotine and capsaicin, four to capsaicin alone, and 22 to nicotine alone. (**h**) Representative traces of hippocampal neurons responding to both nicotine and capsaicin. (**i**) LTP in the Schaffer collateral pathway of TRPV1 KOs is restored to WT levels by 5 μM nicotine exposure 20 min before and during recording ($n = 10$ slices each for WT, TRPV1 KO, WT + nicotine, and TRPV1 KO + nicotine; significance determined by Student's $t$-test, error = s.e.m.). (**j**) The rescue of LTP in TRPV1 KOs by nicotine is not mediated by α4β2 or α7-containing nicotinic receptors—blocked by 100 nM DHBE and 100 nM MLA, respectively—but is mediated by α2 receptors blocked by addition of 1 μM MEC, which blocks all nicotinic receptors including α2 ($n = 7$ slices each for TRPV1 KO + nicotine/DHBE/MLA and for TRPV1 KO + nicotine/DHBE/MLA/MEC; error = s.e.m.).

defects. Most highly nicotine-sensitive nicotinic acetylcholine receptors contain the β2-subunit (CHRNB2). We therefore first examined the localization of TRPV1 and CHRNB2 in OLM neurons. We used a knock-in mouse strain in which all β2-subunits contain an EGFP moiety fused in frame to the β2-subunit[46]. We found that indeed, TRPV1 co-localized with cells expressing CHRNB2 subunits and both somatostatin (Fig. 9d) and reelin (Fig. 9e) in OLM neurons in the stratum oriens. Quantitation indicated that 89.5% of CHRNB2-positive cells in the stratum oriens also expressed TRPV1; CHRNB2-expressing cells were also largely positive for somatostatin and reelin (Fig. 9f). We further tested if TRPV1-expressing hippocampal neurons (that is, neurons responding to capsaicin), also responded to nicotine in $Ca^{2+}$ imaging experiments (Fig. 9g). We found that, indeed, 80% of neurons that responded to capsaicin also responded to nicotine (Fig. 9h). We then tested if acute application of nicotine to hippocampal slices to activate OLM neurons would rescue LTP deficits in TRPV1 knockouts by bypassing the excitatory innervation defects of OLM neurons. Treatment of WT hippocampal slices with 5 μM nicotine before and during LTP recordings caused a mild increase in LTP, as previously reported[45]. While treatment of TRPV1 knockout hippocampal slices with 5 μM nicotine before and during LTP recordings indeed rescued LTP to levels similar to WT slices with nicotine (Fig. 9i).

Nicotinic receptors are present on almost all interneuron subtypes in the hippocampus[47–50], each of which could mediate rescue of LTP in TRPV1 knockouts. For example, intermediate (presumably bistratified) interneurons innervated by OLM neurons[35] may respond to nicotine, and VIP neurons that control OLM cells[51] are also strongly responsive to nicotine[52].

Nicotinic receptor subunits α2–5, and β2–4 have been found in the hippocampus[53], of which α4β2, α7 and α2 subunits are present functionally[47]. α2 is specifically expressed in OLM cells[35,54], has a slow non-desensitizing nicotinic response—unlike α7 and α4 (ref. 47), and is necessary for the gating of LTP[45]. We therefore hypothesized that rescue of LTP in TRPV1 knockouts by nicotine is mediated specifically by α2 receptors on OLM cells. To test this, we isolated α2 from α4β2 and α7 pharmacologically. Application of 100 nM MLA and 100 nM DHBE to block α7 and α4β2 receptors, respectively[52,55], did not block rescue of LTP in TRPV1 knockouts by nicotine (Fig. 9j)—indicating that rescue was indeed mediated specifically via α2 receptors in OLM neurons. To confirm the role of α2 receptors we then added 1 μM MEC, a non-selective nicotinic agonist[47], which blocked rescue of LTP in TRPV1 knockouts by nicotine (Fig. 9j).

We propose a mechanism by which TRPV1 exerts a chronic effect on excitatory innervation of OLM cells to affect plasticity, rather than an acute effect of TRPV1 channel activation during LTP. If this is the case one would expect no effect of acute blockade or activation of TRPV1 on LTP. To test this, we added 1 μM capsaicin or 1 μM SB-366791 30 min before induction of LTP to activate or block TRPV1 channels, respectively. We found no significant effects of capsaicin or SB-366791 on LTP compared to control (Supplementary Fig. 7), consistent with our hypothesis that TRPV1-mediated effects on LTP result from innervation of OLM cells.

Together, our results demonstrate that post-synaptic TRPV1 in OLM cells promotes excitatory innervation of these cells to affect hippocampal plasticity.

## Discussion

We discovered that TRPV1 is expressed in, and promotes excitatory innervation of, OLM interneurons in the hippocampus, which are essential for the gating of information flow in the hippocampus. TRPV1-positive cells expressed somatostatin, a common marker for OLM neurons[25,56], and also expressed reelin. Only 20% of TRPV1-expressing cells were parvalbumin-positive. Interestingly, a recent study found that reelin and somatostatin intersectional expression is restricted to OLM cells, of which there are at least two types: both express somatostatin and mGluR1, but one expresses parvalbumin[31] and originates from the medial ganglionic eminence while the other originates from the caudal ganglionic eminence and expresses reelin and the serotonin receptor 5-HT$_{3A}$R[28]. Our data suggest that TRPV1-expressing cells belong predominantly to the latter sub-population of OLM neurons. We found that these cells also express β2* nAChRs, probably α2β2 (ref. 35), and capsaicin-responsive neurons also responded to nicotine in $Ca^{2+}$-imaging experiments. A previous study failed to detect $Ca^{2+}$ influx using a 10-fold higher capsaicin concentration[10], but this study focused only on neurons of the dentate gyrus and therefore may have missed responses of TRPV1-expressing cells in the stratum oriens.

Perhaps the most striking discovery in our study was that post-synaptic TRPV1 in OLM neurons promotes the formation of exuberant excitatory synapses—presumably from CA1 pyramidal neurons[57,58]—onto these neurons. The functional state of the channel was important since capsaicin increased excitatory synapse number in a dose-dependent manner, and blockade or knockout of TRPV1 dramatically decreased the number of excitatory synapses on OLM neurons. Moreover, heterologous TRPV1 expression in cells that do not normally express it, increased the strength and number of excitatory synapses onto these neurons. To our knowledge, our study is the first to reveal how innervation of OLM neurons is regulated. A recent study reported that ElfN1 recruits mGluR7 receptors to excitatory terminals contacting OLM cells, and decreases glutamatergic release probability—but without affecting the number of excitatory synapses on OLM neurons[59]. TRPV1 then emerges as a candidate molecule that regulates excitatory synaptic strength by modulating glutamatergic innervation of OLM neurons. At higher capsaicin levels (1 μM) synaptic vesicle recycling in excitatory terminals was reduced in both TRPV1-expressing and non-TRPV1-expressing neurons. This is similar to reported effects of acute capsaicin application, which reduced presynaptic excitatory strength[14,15,60]. TRPV1 did not significantly affect inhibitory synapse number or strength. Expression of TRPV1 in neurons normally lacking it; however, decreased inhibitory presynaptic strength in contacting terminals—similar to a recent report that capsaicin depresses GABAergic synaptic transmission in the dentate gyrus[61], without changing inhibitory synapse number.

We propose a pathway whereby NGF upregulates TRPV1 in OLM neurons, which, when activated, leads to increased calcium influx. This activates BDNF expression, which then promotes excitatory innervation of these cells. This pathway is supported by our findings that TRPV1-expressing neurons have higher NGF levels than non-TRPV1-expressing neurons, and that NGF upregulates TRPV1 in hippocampal cultures. This is similar to the peripheral nervous system, where NGF upregulates TRPV1 in nociceptors[33,34]. In the hippocampus, NGF is retrogradely trafficked to septal cholinergic neurons—which innervate OLM cells[35]—and promotes their survival[36]. Thus post-synaptic TRPV1 may be important for cholinergic, as well as glutamatergic innervation. Interestingly, exogenous NGF increases Schaffer collateral LTP, and blockade of endogenous NGF reduces LTP[62], similar to the effects of increasing or blocking TRPV1 function.

A number of endogenous ligands may activate TRPV1 to increase excitatory innervation of OLM neurons. Several endovanilloids have been found to activate TRPV1 in the hippocampus,

including 12-HPETE[14], anandamide[15,63] and N-arachydonoyl-dopamine (NADA)[64]. 2-arachidonylglicerol (2-AG) and N-oleoyl-ethanolamide are also present in the hippocampus and can activate TRPV1. Many of these endovanilloids can activate receptors besides TRPV1. Anandamide and 2-AG, for example, are classified as endocannabinoids because they activate the cannabinoid receptor 1 (CB1) in addition to TRPV1 (refs 15,60,65). Because TRPV1 ligands can activate other receptors, and TRPV1 can be activated by multiple ligands it is possible that more than one endogenous ligand promotes excitatory innervation of OLM cells, potentially at different times, or in response to distinct stimuli.

What are the circuit consequences of reduced excitatory innervation of OLM cells in TRPV1 knockouts? OLM neurons control information flow in the hippocampus by directly inhibiting distal dendrites of CA1 neurons in the temporoam-monic pathway from the entorhinal cortex, and disinhibiting the Schaffer collateral pathway via connections to feedforward inhi-bitory interneurons in the stratum radiatum[35,56]. A reduction in excitatory innervation of OLM neurons would therefore be expected to decrease disinhibition, and subsequently decrease LTP, in the Schaffer collateral pathway. Indeed, we found a reduction in LTP in TRPV1 knockouts in the Schaffer collateral pathway, as previously reported[11,22]. Because OLM neurons directly inhibit the temporoammonic branch of the perforant pathway from the entorhinal cortex[35], TRPV1 knockouts would also be expected to have increased temporoammonic LTP from the perforant path. Interestingly, TRPV1 knockouts do have increased LTP in the medial perforant pathway of the dentate gyrus[15]. However, the medial perforant pathway is presumably not affected by OLM cells. The increased LTP in this pathway in TRPV1 knockouts might rather be due to TRPV1-mediated LTD at the medial perforant path—dentate granular cell synapse[15].

We hypothesized that if the deficit in LTP in TRPV1 knockouts is due to decreased excitatory innervation of OLM neurons, bypassing this innervation defect by activating OLM neurons should rescue LTP in TRPV1 knockouts. Application of nicotine completely rescued the LTP deficit in TRPV1 KOs to WT levels via the α2 nicotinic receptor subunit, which is specifically expressed in OLM neurons[35,54]. This confirms that the LTP deficit observed in TRPV1 knockouts is likely due to decreased excitatory innervation of OLM neurons. In agreement with this hypothesis, we found no effect of acute application of capsaicin or the TRPV1 channel blocker SB-366791 on LTP. Previous studies also report no effect of SB-366791 (ref. 12), or the TRPV1 antagonist capsazepine[13] on LTP. However, these studies do report an increase in LTP with addition of capsaicin[12,13]; a 1,000-fold higher concentration of capsaicin was used in the former publication, and a theta burst protocol was used in the latter. It is therefore likely that the concentration of capsaicin or induction protocol may affect the degree of potentiation of LTP by capsaicin.

Gibson et al.[14], described a TRPV1-mediated depression of stratum radiatum interneuron responses during high-frequency stimulation of the Schaffer collateral, which was absent in TRPV1 knockouts. This would result in excess inhibition of pyramidal neurons and reduced Schaffer collateral LTP in TRPV1 knockouts (which we, and others have observed[11,22]). Our model is largely in agreement with Gibson et al., except for where TRPV1 is located. We both agree that TRPV1 affects stratum radiatum interneurons, but is not present in stratum radiatum interneurons themselves–based on immunostaining in our case, and absence of blockade of depression by introducing TRPV1 blockers into recorded stratum radiatum interneurons in Gibson et al. In terms of TRPV1 location, however, Gibson et al. proposed that TRPV1 is presynaptic on pyramidal neurons, where it reduces transmitter release onto post-synaptic stratum

radiatum interneurons during high-frequency stimulation of the Schaffer collateral. We propose that TRPV1 is present in OLM neurons that contact stratum radiatum interneurons, where TRPV1 promotes excitatory innervation of OLM neurons. This would also act to depress stratum radiatum interneurons (via disinhibition) during stimulation of the Schaffer collateral pathway. We favour this model because we observed capsaicin responses and TRPV1 staining specifically in OLM neurons (and not in TRPV1 knockouts), and because specific activation of OLM neurons by α2-nicotinic receptors-rescued LTP in TRPV1 knockouts. However, it remains possible that both pathways could work in parallel.

The presence of TRPV1 in OLM neurons also may explain the observation that OLM neurons become especially hyper-excitable at febrile temperatures (40–43 °C) (ref. 66). TRPV1 could participate in the aetiology of febrile seizures. In animals with persistent epileptic seizures the firing frequency of OLM interneurons shifts from normal theta frequency[17,25] to gamma oscillations, possibly due to enhanced frequency of excitatory input[67]. This, in turn changes the excitability of the entire neuronal network[68]. OLM cells can therefore have profound effects on the total excitatory and inhibitory output of the hippocampal system. Interestingly, a subset of OLM cells with high excitation are recruited during hippocampal sharp wave ripple events[69]. These events are important for memory consolidation and reactivation of place cells[70]. TRPV1 may be involved in sharp wave ripples—by promoting high excitatory innervation of this subset of OLM neurons—and therefore important for intrahippocampal information processing necessary for spatial memory.

## Methods

**Animals.** Use of animals for experimentation, including TRPV1 knockout[1] (B6.129X1-Trpv1$^{tm1Jul}$/J), PV-Tomato (B6;129P2-Pvalb$^{tm1(cre)Arbr}$/J × B6;129S6-Gt(ROSA)26Sor$^{tm9(CAG-tdTomato)Hze}$/J) and VIP-Tomato (Vip$^{tm1(cre)Zjh}$/J × B6;129S6-Gt(ROSA)26Sor$^{tm9(CAG-tdTomato)Hze}$/J) reporter mice, all acquired from The Jackson Laboratory, was approved and performed according to the specifications of the Institutional Animal Care and Ethics Committees of Göttingen University (T10.31 and T10.24), and of the German animal welfare laws. Use of β2-EGFP nAChR mice[46] was approved by the Institutional Animal Care and Ethics Committee at Lehigh University (protocol number 147).

**Plasmids and expression constructs.** In initial experiments we used TRPV1-IRES-EGFP (provided by Gero Miesenboeck) compared to cytosolic enhanced green fluorescent protein (EGFP) to test the effects of TRPV1 expression on synapse formation and function. In subsequent experiments, we generated new plasmids TRPV1-P2A-EGFP and VR.5′ sv-P2A-RFP using the chicken β-actin promoter to drive expression of TRPV1 and VR.5′ sv proteins co-expressing EGFP (with TRPV1) or RFP (with VR.5′ sv) using the P2A multicistronic element. The β-actin promoter-driven TRPV1-P2A-EGFP and VR.5′ sv-P2A-RFP plasmids were sub-cloned by GenScript (Piscataway, NJ, USA). Synapsin promoter-driven EGFP and ChR2-EYFP AAV vectors were provided by Karl Deisseroth (Stanford University).

**Antibodies and reagents.** Primary antibodies used were: chick and rabbit EGFP (1:4,000 dilution; Abcam), mouse GAD65 (1:1,000), guinea pig (1:800 for uptake) and mouse syt1 (1:150 for uptake), rabbit alpha tubulin (1:5,000), mouse syb2 (1:1,000 dilution; Synaptic Systems), chick and mouse MAP2 (1:4,000 dilution; Millipore), sheep NGF, mouse reelin, guinea pig TRPV1 C-T (1:1,000, Millipore), mouse TRPV1 N-T (1:500; Neuromab), rabbit TRPV2, TRPV3, TRPV4 (1:500; Alomone), rabbit mGluR7 (1:500; Upstate), mouse PV25 (1:2,000 dilution; Swant), rabbit Somatostatin-14 (1:2,000 dilution; Bachem), rat Somatostatin-14 (1:1,000 dilution; Thermo Fisher), rabbit VIP (1:500 dilution; Abcam), mouse BDNF (1:500; Developmental Studies Hybridoma Bank; Yves Barde monoclonal #9). Secondary antibodies used were: guinea pig, mouse and rabbit HRP (1:5,000 dilution; BioRad) and chick, rabbit, guinea pig, mouse, rat and sheep Alexa 405, 488, 546 and 647 (1:2,000 dilution; Abcam/Invitrogen). Neurotrace was from Thermo Fisher.

Capsaicin and SB-366791 were purchased from Sigma-Aldrich. Methyllycaconitine (MLA) was from Abcam and dihydro-beta-erythroidine (DHBE), mecamylamine (MEC) picrotoxin, TTX, APV and CNQX were from Tocris. Recombinant Human TrkA Fc and TrkB Fc Chimera Proteins were from R & D Systems. All other reagents were from Carl Roth.

**Dissociated rat hippocampal neuron preparation.** E18 pregnant Wistar rats were killed with $CO_2$, embryos were removed, and heads of embryos were placed in a 10 cm petri dish containing ice-cold dissection media (HBSS (Gibco) + 10 mM Hepes (Gibco)). Brains were removed and collected in fresh dissection medium. Hippocampi were separated from the brain and meninges were removed. Hippocampi were digested with 2 ml pre-warmed 37 °C 0.05% trypsin-EDTA (Gibco) for 20 min at 37 °C. Trypsin was removed and the tissue was washed three times with 4 °C dissection medium. Dissection medium was replaced with 1 ml pre-warmed NB+ (Neurobasal with 1X B-27 supplement, 1X Glutamax and Penicillin (5,000 U ml$^{-1}$)/Streptomycin (5000 µg ml$^{-1}$); all from Gibco) and tissue was triturated by gentle pipetting. The tissue suspension was filtered through a 100 µm cell strainer (BD Biosciences). Cells were counted using the trypan blue exclusion method and cultured on 12 mm glass coverslips (Thermo Scientific) coated with poly-D-lysine (PDL, Sigma) dissolved in 0.1 M borate buffer, in 24-well plates (CytoOne) at a density of 80,000 hippocampal neurons per cm$^2$ in NB+ medium in a Hera Cell 240i cell culture incubator (Thermo Scientific) at 37 °C and 5% $CO_2$.

**Dissociated mouse hippocampal neuron preparation.** TRPV1 knockout mice[1] were purchased from Jackson Laboratory. P0 TRPV1 knockout and WT mice of both sexes were dissected successively; pups were decapitated and heads placed on a petri dish and sprayed with 70% ethanol. The skin of the scalp was separated, skull was opened, and the brain was removed and placed in a petri dish containing 4 °C dissection medium. Hippocampal dissection was performed as for rat embryonic hippocampal dissection with a few modifications. Hippocampi were digested with either 2 ml pre-warmed 37 °C 0.05% trypsin-EDTA (Gibco) for 20 min at 37 °C, or filtered papain-based enzymatic solution (11.39 mM L-cysteine, 50 mM Na-EDTA (pH = 8), 100 mM CaCl$_2$, 1 N NaOH, 100 µl papain (Worthington) (added to the solution 15 min before use) and 10 mg ml$^{-1}$ of DNAseI (added just before digestion) for 30 min at 37 °C. For trypsin digestion, trypsin was removed and the tissue was washed three times with 4 °C dissection medium. Dissection medium was replaced with 1 ml pre-warmed NB+ (Neurobasal with 1X B-27 supplement, 1X Glutamax and Penicillin (5,000 U ml$^{-1}$)/Streptomycin (5,000 µg ml$^{-1}$); all from Gibco) and tissue was triturated by gentle pipetting. For papain digestion, the enzymatic solution was removed and replaced by an inactivation solution (BSA, 5% serum medium and 10 mg ml$^{-1}$ of DNAseI) at 37 °C for 15 min. Subsequently the tissue was triturated and then centrifuged in an Eppendorf 5810-12 centrifuge at 500 g, the supernatant was discarded and replaced with fresh NB+, and the tissue pellet was resuspended before plating neurons at a density of 120,000 cells per cm$^2$.

**Transfection of dissociated rat hippocampal neurons.** To achieve low-efficiency heterologous expression of TRPV1 and VR.5′ sv in single neurons, lipofectamine-based transfections were performed in cultured rat or mouse hippocampal neurons at DIV2. For each well of a 24-well plate, 0.75 µg of plasmid DNA and 1 µl of the cationic lipid-based transfection reagent Lipofectamine 2000 (Invitrogen). were each diluted in 50 µl of 4 °C Neurobasal in separate tubes and incubated for 5 min at room temperature (RT). Subsequently, both dilutions were mixed together and incubated at RT for 20 min. During this time, the culture medium from each well to be transfected was exchanged with 400 µl of new pre-warmed NB+ (Neurobasal with 1X B-27 supplement, 1X Glutamax and penicillin (5,000 U ml$^{-1}$)/streptomycin (5,000 µg ml$^{-1}$); all from Gibco) at 37 °C; the original culture medium was stored at 37 °C and 5% $CO_2$. After incubation, 100 µl of the lipofectamine-DNA mix was added to each well for a final volume of 500 µl per well. Cultures were incubated for 2 h at 37 °C and 5% $CO_2$. Transfection medium was then removed, and wells were washed once with 500 µl Neurobasal at 37 °C, and 450 µl of the original culture medium at 37 °C was added back to the wells. Reporter gene expression was normally detected 2 days after transfection.

**ChR2-EYFP AAV production, transduction and LED stimulation.** Recombinant AAV serotype 1/2 particles were generated by transfecting 60% confluent HEK293 cells in 10 cm dishes (Greiner Bio-One) cultured at 37 °C and 5% $CO_2$ in 10 ml of medium consisting of DMEM, 10% fetal bovine serum, 100 U ml$^{-1}$ penicillin and 100 µg ml$^{-1}$ streptomycin (all from Gibco), using calcium phosphate transfection. For one 10 cm dish, 150 µl 2 M CaCl$_2$, 80 µg plasmid DNA (20 µg EGFP or ChR2-EYFP expression vector, 40 µg adenovirus helper plasmid pFΔ6 and 10 µg each of AAV1 (pH21) and AAV2 (pRV1) replication and capsid sequence plasmids), and dH$_2$O to a final volume of 1,200 µl was mixed dropwise with 1,200 µl transfection buffer (in mM: 274 NaCl, 10 KCl, 1.4 Na$_2$HPO$_4$, 15 glucose, 42 HEPES; ph = 7.05–7.12, sterile filtered). This mixture was incubated for 20 min at room temperature, and then added dropwise to 10 cm dishes of HEK293 cells at 60% confluence. 48–72 h post transfection, cells were harvested in 500 µl DPBS and lysed by four repeated freezing (−80 °C) and thawing (37 °C) cycles. Cell debris was removed by centrifuging at 10,000 r.c.f. for 5 min. The supernatant, containing crude AAV extract was collected and stored at 4 °C. For viral transduction, 1–2 µl of this supernatant was applied per well of six-well plates of dissociated hippocampal cultures at DIV3, to achieve a transduction efficiency of >80% by DIV13. On DIV13, EGFP or ChR2-EYFP transduced neurons were transferred to a custom-built LED chamber containing 100 mW blue LEDs (Conrad) operated at

3.2 V with an output of 0.1 mW mm$^{-2}$, placed beneath a polyoxymethylene light diffuser (Kern GmbH) 20 mm below the cells. A micro-controller connected to a power supply delivered three light pulses of 100 ms duration every 1.5 s to cultures in the incubator for 21–24 h. Following stimulation, cultures were fixed, immunostained for EGFP or EYFP, TRPV1, vGluT1 and MAP2, and number of synapses quantified as below.

**Synaptotagmin 1 luminal domain antibody uptake.** For synaptotagmin-1 luminal domain antibody uptake (syt1u) dissociated hippocampal neurons were incubated for 10 min at 37 °C in 250 µl high potassium depolarizing buffer (75 mM NaCl, 70 mM KCl, 20 mM Hepes, 2 mM CaCl$_2$, 5,5 mM glucose 2 mM MgCl$_2$, and 20 mM Hepes; pH 7.3) containing syt 1 luminal domain specific antibodies. Depolarizing buffer was then removed and samples were fixed for immunostaining.

**Immunocytochemistry.** Dissociated hippocampal cultures growing on glass coverslips were fixed with 250 µl 4% paraformaldehyde (PFA) diluted in 0.1 M phosphate buffer (PB) (0.1 M Na$_2$HPO$_4$, 0.1 M NaH$_2$PO$_4$ dissolved in ddH$_2$O; pH 7.4) for 20–30 min at room temperature. Fixed hippocampal cultures were then washed three times with 500 µl 1 × PBS (and stored up to one month at 4 °C in 1 × PBS). Dissociated cultures were then incubated for 20 min at room temperature in 250 µl blocking buffer containing Triton X-100 for permeabilization (2% [v/v] donkey serum, 0,1% (v/v) Triton X-100 (Carl Roth), 0,05% (v/v) NaN$_3$ in 2X PBS). Samples were then incubated with primary antibodies in blocking buffer overnight at 4 °C. The solution containing primary antibodies was discarded and cells were washed three times for 5 min each with 500 µl 1 × PBS. Cells were then incubated for 2 h at room temperature in the dark in 250 µl blocking buffer containing secondary antibodies conjugated to a fluorophore. Secondary antibody solution was then removed and cells were washed three times for 5 min each with 1 × PBS. For DAPI staining samples were incubated for 10 min in 250 µl 1 × PBS containing 10 µM DAPI (Invitrogen) at room temperature in darkness and washed three times for 5 min each with 1 × PBS. The same primary or secondary antibody mix was used for different conditions or treatments within the same experiment. Coverslips were mounted with a drop (10 µl) of Fluoromount Plus (Diagnostic Biosystems) or Fluoromount-G (Sigma) on glass slides (Carl Roth). Coverslips were sealed using transparent nail polish, and then imaged by epifluorescence and confocal microscopy.

**Immunohistochemistry and paraformaldehyde perfusion fixation.** Two- to three-month old WT (B6/J), TRPV1 KO, PV-Tomato, VIP-Tomato and 3–5-month-old β2-EGFP nAChR mice of both sexes were anaesthetized by intraperitoneal injection of a tranquilizer cocktail containing 10% ketamine, 2% xylazine and 0.9% saline solution or pentabarbitol solution for the β2-EGFP nAChR mice (0.1 ml per 10 mg body weight). After anaesthetization mice were placed on a perfusion chamber (abdomen side up) and fastened with needles. The rib cage was removed and once the heart was successfully exposed, the right atrium was cut and a winged needle connected to a peristaltic pump (Masterflex) was inserted into the left ventricle. The blood was drained by perfusion of pre-filtered sucrose solution (10% sucrose (Carl Roth) in ddH$_2$O) for ~1 min followed by ~50 ml of pre-filtered 4% PFA at RT for ~20 min (for fixation). A successful perfusion was detected when the tail of the animal became stiff. The brain was subsequently dissected out of the skull, incubated in sterile-filtered 4% PFA at 4 °C for 2 h, and washed twice for 15 min each with 0.1 M PB. Tissue was then used for IHC or stored in PB at 4 °C for up to a month. All buffers and solutions were freshly prepared for every perfusion. For the β2-EGFP nAChR mice, the same procedure was used as above, except that a PBS wash was performed before 4% PFA perfusion, and brains were post fixed in 4% PFA o/n at RT and then switched to 1% PFA solution for storage.

**Immunohistochemistry of mouse brain sections.** Before tissue sectioning, 2–3-month-old fixed mouse brains were pre-sectioned using an acrylic brain blocker (Kopf Instruments) to remove the most frontal and caudal parts of the brain to keep the starting point for slicing similar for every brain. Coronal brain sections were generated using a Leica VT 1200 vibratome (Leica Biosystems): Pre-sectioned brains were stabilized caudally with Roti coll 1 superglue (Carl Roth) on a specimen disc fixed to the vibratome's buffer tray. The tray was then filled with 4 °C 0.1 M PB. The system was calibrated and 40-µm-thick coronal sections were cut using a razor blade (Astra Platinum). Each slice was collected in individual wells of a 24-well plate containing 4 °C 0.1 M PB and stored up to a month at 4 °C. Brain sections were visually selected for immunohistochemistry using an Axiovert 40 CFL (Zeiss). Comparable sections between different brains from different samples were sorted and classified depending on their approximate stereotactic coordinates according to Paxinos and Franklin[71]. Sections were washed first with 50 mM Tris buffer (TB) (38.5 mM Tris-HCl; 11.5 mM in ddH$_2$O; pH 7.6), then with Tris-buffered saline (TBS) (0.9% NaCl in TB, pH 7.6) and finally with Tris-buffered saline with Triton X-100 (TBST) (0.5% [v/v] 100X Triton X-100 in TBS; pH 7.6). Every washing step was done twice for 15 min each at RT. sections were then incubated for 90 min in blocking buffer, and then in 500 µl blocking buffer containing primary antibodies for 48–72 h at 4 °C on an orbital shaker.

Subsequently, tissue was washed $4 \times 15$ min with TBST at RT. Then tissue was incubated for 2 h at RT in 500 μl blocking buffer containing secondary antibodies in darkness. Tissue was then washed with TBST ($2 \times 15$ min) and with TBS ($1 \times 15$ min) at RT. For DAPI staining the tissue was incubated for 10 min in 500 μl $1 \times$ PBS containing 10 μM DAPI (Invitrogen) at RT in darkness. Tissue was then washed at RT once for 15 min with TBS and then two times for 15 min each with TB. Brain sections were mounted on glass slides (Carl Roth) with a drop (10 μl) of Fluoromount-G (Sigma) and covered with 12 mm glass coverslips (Thermo Scientific). Coverslips were sealed using transparent nail polish and imaged by confocal microscopy.

**Microscopy.** Immunostained dissociated hippocampal neuronal cultures on glass coverslips were imaged by epifluorescence using an upright Zeiss Axiovert.A1 microscope (Carl Zeiss) with a $\times 100/1.4$ oil Plan-Apochromat objective or a $\times 63/1.4$ oil Plan-Apochromat objective (Carl Zeiss). Images were captured with a Retiga-SRV camera (QImaging) and QCapture software (QImaging). Three to ten pictures of different neurons/regions of a given condition within the same experiment were acquired using identical settings for image size, light intensity, gain and exposure time. Confocal microscopy was conducted using a Zeiss LSM 710 confocal microscope (Carl Zeiss) with a $\times 40$ Plan-Neofluar oil immersion objective or $\times 20$ Plan-Apochromat air objective. A diode laser was used to excite the Alexa 405 fluorophore (excitation–emission: 401–421 nm) and DAPI (emission: 410–495 nm). The Alexa 488 fluorophore (excitation–emission: 493–520 nm) was excited with a 488 nm argon laser. The Alexa 546 fluorophore was excited with a DPSSL laser (excitation–emission: 556–573 nm) and the Alexa 647 fluorophore was excited with a HeNe laser (excitation–emission: 650–665 nm). A minimum of three images of hippocampal sections or neurons within the same experiment were acquired with identical frame size, scan speed, zoom, laser intensity, master gain, pinhole airy units, digital offset and digital gain. Images were captured and saved using ZEN 2009 software and exported as 8-bit RGB tiff files.

**Synaptic strength and synapse number quantitation.** To optically measure synaptic strength of hippocampal neurons *in vitro* we co-immunostained for syt1 uptake to mark recycling synaptic vesicles and vGluT1, to mark synaptic vesicles at excitatory presynaptic terminals, or vGAT, to mark synaptic vesicles at inhibitory presynaptic terminals (or with BDNF in experiments comparing BDNF puncta intensity on WT and TRPV1 knockout neurons). The intensity of vGluT1 or vGAT signal corresponds to the number of synaptic vesicles per terminal and, in conjunction with syt1 uptake, gives a readout of presynaptic strength.

A mask of the transfected cell was created according to GFP or RFP fluorescence for transfected neurons, or endogenous TRPV1 or SOM/reelin immunostain signal, using Adobe Photoshop CS5 software (Adobe Systems Incorporated). All synaptic puncta positive for vGluT1 or vGAT on single transfected neurons within the mask were selected using OpenView 3.0 software (provided by Noam Ziv, Technion, Israel Institute of Technology, Haifa, Israel). The fluorescent intensity of vGluT1 or vGAT signal was measured using the 'place area over puncta' function; threshold and delta intensity were configured for an accurate selection of synapses. Fluorescence intensity of puncta within the transfected neuron mask reported presynaptic strength on the selected neuron. The corresponding fluorescence intensity of the co-stained syt1 antibody taken up by recycling synaptic vesicles within selected areas was also measured. Blurry regions, puncta on somas and puncta not overlapping a visible neuronal process were excluded from analysis.

The values for fluorescence intensity generated by OpenView 3.0 were imported into Microsoft Excel. To compare conditions in multiple experiments, the fluorescence intensities in different conditions within a single experiment (a neuronal cell preparation in which ICC for all conditions was performed using the same antibody mix) were normalized to a range from 1 to 100, where 1 corresponded to the lowest and 100 to the highest fluorescence intensity detected in the experiment using the formula:

$$x = \frac{i - \min}{\max - \min} \times 100,$$

where $x =$ normalized value, $i =$ intensity value calculated by OpenView 3.0, $\min =$ lowest intensity value of the experiment and $\max =$ highest intensity value of the experiment. The fluorescence intensity of puncta in every image was averaged and the mean of the average fluorescent intensity across all experiments for multiple images was calculated for each condition. Pooled data of all images for every condition were plotted with the standard error of the mean. BDNF puncta intensity on WT and TRPV1 knockout neurons was determined using a similar method by thresholding puncta using OpenView 3.0 software.

Synapse number (or BDNF puncta number) was analysed by visual selection and counting of puncta positive for vGluT1 or vGAT (or BDNF) immunofluorescence overlapping EGFP or MAP2—a marker of dendrites in mature neurons. Synapses were counted along 2–3 randomly selected 30–50 μm long segments of proximal dendritic processes per neuron. Since TRPV1-expressing neurons have dense innervation where synapses are tightly packed, making the identification of individual synapses difficult, MetaMorph (Molecular Devices) software was used to calculate the number of synapses contacting these neurons. The fluorescent signal of vGluT1 and vGAT-positive immunostainings

was selected by thresholding the signal (using the 'set colour threshold' function) to a specific level where only synapses were selected. Then, the 'threshold' intensity value of a randomly selected 30–50 μm length of a proximal dendrite was used to calculate the number of synapses in this area using the following formula:

$$n = \frac{i/d}{l/p},$$

where $n =$ number of synapses, $i =$ average fluorescence intensity in a selected thresholded area, $d =$ average area of a single synapse (in pixels), $l =$ length of the sample area (in pixels) and $p =$ number of pixels per micron.

Synapse number for every condition was then normalized to control conditions (EGFP transfected, or untreated condition). The normalized synapse number of every image was averaged and the mean of the average synapse number across all experiments was calculated for each condition. Pooled data from all images for every condition was plotted with the standard error of the mean.

**Somatic protein expression quantitation.** To assess the somatic expression level of specific proteins (for example, TRPV1, NGF and BDNF), the average fluorescence intensity in regions placed around somas was measured using ImageJ or MetaMorph software. For quantitation of TRPV1 expression, the signal in surrounding non-expressing TRPV1 neurons was set to zero, and any signal remaining in WT (or TRPV1 knockout) neurons was calculated relative to zero. The average soma fluorescence intensity across all experiments was calculated for each condition. Pooled data from all images for every condition were plotted with the standard error of the mean.

**Calcium imaging.** To test if capsaicin activates hippocampal neurons *in vitro* we used the Fluo-4 (Invitrogen) calcium indicator at a concentration of 1:1,000 added to DIV 13–14 mouse hippocampal neurons. After 10–20 min of incubation, coverslips were placed on a field stimulation imaging chamber (Warner Instruments) in 100–150 μl of base solution (140 mM NaCl, 5 mM KCl, 2 mM CaCl$_2$, 2 mM MgCl$_2$, 5.5 mM glucose, 20 mM Hepes; pH $= 7$). For experiments in which blockade of TRPV1 was tested, the base solution contained 10 μM APV, 10 μM CNQX and 1 μM TTX, with or without 1 μM SB-366791. The stimulation chamber was connected to a biphasic stimulator and positioned on a Zeiss AxioObserver A1 epifluorescence microscope (Carl Zeiss) with a DG-4 wavelength switcher (Sutter Instruments). A $40 \times /1.4$ air Plan-Apochromat objective and a 488 nm wavelength filter was used to detect Fluo-4 fluorescence (excitation–emission: 494–516 nm). Three-minute time series of 1 frame per second were acquired with an exposure time $\leqslant 200$ ms using an Evolve Camera (Photometrics) controlled by MetaMorph software (Molecular Devices). After 1 min baseline recording, 30 pulses of 10 V at 40 Hz and 1 ms duration were delivered by field stimulation. Following field stimulation, a final concentration of 1 μM capsaicin, or 5 μM nicotine followed by 1 μM capsaicin (or corresponding volume of DMSO) was added by pipette and samples were imaged for an additional minute.

For analysis of Ca$^{2+}$ responses, somas of neurons responding to field stimulation were selected with the region tool in MetaMorph. Background (the average intensity of five regions without visible neuronal processes or cell bodies) was subtracted and the average intensity in cell somas throughout the time course was determined using Metamorph and further analysed using a custom written script in Matlab. Frames were clipped from individual traces such that time of field stimulation and time of addition of capsaicin or DMSO in each trace was aligned with respect to each other. Fluorescence averaged over the first 40 frames was taken as baseline ($F_0$). The $\Delta F/F_0$ traces were calculated over time ($t$). Cells that had $< 20\%$ increase upon field stimulation were excluded from analysis for comparison of WT and TRPV1 knockout calcium responses. Cells that showed $> 10\%$ increase in Fluo-4 fluorescence upon capsaicin or nicotine addition, were considered responsive.

**RNA isolation and RT–PCR.** To detect expression of TRPV1 and its isoforms, RNA was isolated from hippocampus and dorsal root ganglia (DRG) from 2- to 3-month-old WT (B6/J) and TRPV1 KO mice of both sexes. Animals were killed with CO$_2$ and hippocampi and DRG were dissected and isolated in sterile $1 \times$ PBS in DNase/RNase free Eppendorf tubes (ThermoFisher), snap-frozen with liquid nitrogen and stored at $-80 °C$. Tissue was dissected from each mouse using separate tools to avoid contamination.

Hippocampi and DRG were homogenized with 500 μl or 250 μl Trizol (Sigma Aldrich), respectively, and incubated at RT for 5 min. Subsequently, 100 μl (hippocampus) or 50 μl (DRG) chloroform was added and mixed vigorously by inverting the tube 15x and allowing solution to settle for 3 min at RT. Samples were then centrifuged for 15 min at $4 °C$ at 13,000 r.c.f. The upper aqueous supernatant was transferred to a new tube, isopropanol was added (250 μl for hippocampus and 125 μl for DRG) and mixed by gentle inversion followed by -20 °C incubation for 1 h. Samples were then centrifuged at 13,000 r.c.f. for 30 min at $4 °C$. The supernatant was discarded and the RNA pellet was washed with 1 ml ice-cold 70% ethanol. After washing, a final centrifugation step of 5 min at $4 °C$ RT 13,000 r.c.f. was performed, supernatant was discarded and the RNA pellet was air-dried at RT. The RNA pellet was dissolved in DNase- and RNase-free water (30 μl for hippocampus and 20 μl for DRG). Samples were treated with DNase I

(Ambion) to eliminate any remaining chromosomal DNA. Purity and concentration of RNA was determined using a spectrophotometer (NanoDrop ND-1000) with ND-1000 v3.5.2 software (Peqlab Biotechnologie). Only isolated RNA with an integrity value of more than 8 was used.

cDNA was synthesized using 2 µg (hippocampus) or 1 µg (DRG) of the isolated RNA: RNA samples were diluted with DNase- and RNase-free water to a final volume of 11 µl and mixed with 9 µl of the cDNA synthesis reaction mix (4 µl 5 × reaction buffer, 0.5 µl RNase inhibitor, 2 µl deoxynucleotides, 0.5 µl reverse transcriptase, 2 µl random hexamer mix from Roche Applied Science) for a final volume of 20 µl. RNA was reversed transcribed with the Transcriptor High Fidelity cDNA synthesis kit (Roche Applied Science) under the following conditions: 1 × 10 min 25 °C; 1 × 30 min 55 °C; and 1 × 5 min 80 °C, followed by a hold at 4 °C. A reverse transcription–PCR (RT–PCR) without reverse transcriptase served as a negative control. Isolated RNA and negative controls were stored at − 20 °C.

qPCR was performed on a light cycler 480 system (Roche Applied Science) using the LC480 SYBR green Master mix (Roche Applied Science). 2 µl of the synthesized cDNA library was used for qPCR with TRPV1 primers (see below) and 1:10 diluted cDNA was used for the reference control tubulin α-1B. All data obtained was normalized to mRNA levels of tubulin alpha-1B. The qPCR reaction mix contained 2 µl cDNA, 0.75 µl 10 µM forward and reverse primers, 7.5 µl LC480 2 × Master mix, and 4 µl DNase- and RNase-free water. Cycling conditions were: 95 °C 5 min denaturation, repeated cycle of 95 °C 10 s, 60 °C 15 s, 72 °C 10 s for 45 ×, one cycle of 95 °C 15 s, 67 °C 30 s, 95 °C 15 s for the melting curve, followed by 40 °C 10 s, and 4 °C hold.

Forward and reverse primers, respectively, were used for identification of the TRPV1 channel pore region; TRPV1 PR (5′-cttgtttggatttttccacagc-3′, 5′-cacttg-tgtggtggggact-3′), the TRPV1 N-terminus; TRPV1 N-T (5′-gagtgtgcctgcacctagc-3′, 5′-tcagattcatccgagtctaagc-3′), the TRPV1 C-terminus; TRPV1 C-T (5′-ggcatgt-ccctaggacttca-3′, 5′-agctcgcttcccacacac-3′), the vanilloid receptor 5′-splice variant; VR.5′sv (5′-tgaccctcttggtggagaatgg-3′, 5′-tgaggctagtccctggatgactg-3′) and for tubulin alpha-1B (5′-aggagctggcaagcatgt-3′, 5′-agctgctcaggctggaagag-3′). All primers were synthesized by and purchased from Sigma-Aldrich.

For VR.5′ sv, the only published sequence to our knowledge was reported for rat (Rattus norvegicus) tissue[21] (GenBank accession number AF158248.2). We designed primers to detect the same region in mouse, by aligning the TRPV1 genomic mouse sequence (GenBank accession number NC_000077.6) with the rat TRPV1 genomic sequence (GenBank accession number NC_005109.4) using the open source software 'A plasmid editor' v2.0.47 (Copyright 2003–2009 by M. Wayne Davis) (ApE). We chose primers using the online tool 'Universal Probe Library Assay Design Center' (Roche Diagnostics) by entering the complete NCBI reference sequence (NM_001001445.2) for TRPV1, the sequences above for VR.5′ sv, or the specific exons we aimed to detect, that is, the amino and carboxyl termini or pore region of TRPV1 or the unique VR.5′ sv sequences.

**Biochemistry and western blotting.** For preparation of whole brain and hippocampus homogenates, tissue was dissected from 2–4-month-old mice of both sexes and collected in DNase- and RNase-free Eppendorf tubes and snap frozen in liquid nitrogen. Tissue was homogenized in 9 ml of 4 °C homogenization buffer (320 mM sucrose and 4 mM HEPES; pH 7.4, adjusted with NaOH; supplemented with protease inhibitors; 1:1,000 dilution) using a glass-teflon homogenizer with 8–10 strokes at 900 r.p.m. Protein concentration was determined by BCA assay (Novagen). Absorbance was measured at 562 nm on a plate reader. Protein separation was performed following the Schägger protocol. Proteins were denatured: samples were mixed with Schägger sample buffer (50 mM Tris 4% SDS, 0.01% Serva Blue G, 12% glycerol; pH 6.8, adjusted with HCl; for reducing 2% mercaptoethanol) and heated at 99 °C for 10 min. Equal amounts of protein were loaded into each lane and proteins were separated by 10% SDS–polyac under constant current using a Bio-rad electrophoresis power supply. Proteins were transferred from a polyacrylamide gel to a nitrocellulose membrane at 100 mA for 1 h.

For immunoblotting, membranes were incubated for 1 h at RT in blocking buffer (5% lyophilized non-fat milk in 1X TBST). Primary antibody was then added to the blocking buffer and the membrane was incubated O/N at 4 °C in primary antibody solution. The membrane was subsequently washed 3 × with TBST and then incubated for 1 h at RT with HRP-conjugated secondary antibodies in blocking buffer. After washing 3 × with TBST, the custom-made chemiluminescent substrate ECL consisting of 1 ml of solution A (2.42 g Tris-HCl, 500 mg luminol sodium salt (Sigma-Aldrich) dissolved in 200 ml ddH₂O), 100 µl of solution B (11 mg P-coumaric acid (Sigma-Aldrich) dissolved in 10 ml DMSO) and 0.3 µl of solution C (30–35% [v/v] of H₂O₂) was poured on the membrane and incubated for at least 3 min in darkness. For visualization, signal was captured using a LAS reader (Fuji). The signal intensity was quantified using the open source software Image J, and values were plotted in Excel (Microsoft).

**mEPSC recordings in dissociated cultures.** Whole-cell patch clamp recordings from cultured rat hippocampal neurons (growing on glass coverslips) were performed at RT at DIV12–14 in a custom-made recording chamber. Transfected neurons were visualized with an upright Olympus BX51WI microscope (Olympus) with 10 × /0.3 water UMPlanFL N and 40 × /0.3 LUMPlanFL N objectives and a Lumen 200 Pro fluorescent light source (Prior Scientific) and filters for GFP

detection to identify transfected neurons. The artificial cerebrospinal fluid (ACSF) extracellular solution contained (in mM) 136 NaCl, 2.5 KCl, 10 HEPES, 10 D-glucose, 2 CaCl₂ and 1.3 MgCl₂; pH 7.4, adjusted with 10 M NaOH; 290–300 mOsm; 1 µM TTX (Tocris). ACSF was bubbled with carbogen (95% O₂ and 5% CO₂) to achieve a pH of 7.2. For mEPSC recordings, the extracellular solution included 100 µM picrotoxin (Tocris); for mIPSC recordings, 10 µM CNQX and 10 µM APV (Tocris) were added to the ACSF. The intracellular solution contained (in mM) 130 K-gluconate, 10 NaCl, 1 EGTA, 0.133 CaCl₂, 2 MgCl₂, 10 HEPES, 3.5 Na-ATP, 1 Na-GTP; pH = 7.3; 280 mOsm) for mEPSC detection and (in mM) 136 CsCl, 10 EGTA, 2 CaCl₂, 0 MgCl₂, 10 HEPES, 2 Mg-ATP, 0.3 Na-GTP; pH = 7.3, adjusted with CsOH; 290–295 mOsm for mIPSC detection. Recordings were performed with a HEKA EPC10 USB double patch clamp amplifier coupled to Patchmaster acquisition software (HEKA Electronics Inc.). Borosilicate glass 2–5 MΩ micropipettes used for recording were pulled using a P-97 micropipette puller (Sutter Instruments). Neurons were held in voltage-clamp mode at − 60 mV for both mEPSC and mIPSC recordings. Data was filtered using a low-pass Bessel filter at 2.9 kHz and digitized at 5 kHz. Recorded traces were analysed using Mini Analysis v6.0.3 software (Synaptosoft). The amplitude threshold for detection of mEPSCs was approximately 3.5 × RMS noise. Only undisturbed traces of at least 1 min. were considered for analysis. Average amplitude and frequency were plotted with standard error of the mean.

**Whole-cell recordings from OLM neurons in hippocampal slices.** For recording mEPSCs and capsaicin responses in OLM neurons from hippocampal slices, P10-P24 mice of both sexes were anaesthetised with isofluorane (Abbott, Wiesbaden, Germany) and decapitated. The brain was dissected and placed in ice-cold carbogen (95% CO₂/5% O₂) -bubbled NMDG cutting buffer (containing in mM: 45 NMDG, 0.3 KCl, 0.4 KH₂PO₄, 0.5 MgCl₂, 0.16 CaCl₂, 20 choline bicarbonate, 12.95 glucose). The cerebellum was removed and the brain glued to a vibratome chamber filled with carbogen-bubbled ice-cold NMDG buffer. 300 µm thick coronal hippocampal slices were cut using a Leica VT1200 vibratome (Wetzlar, Germany) and a stainless steel blade (Feather). Slices were placed in a submerged chamber containing carbogen-bubbled ACSF (in mM: 124 NaCl, 4.4 KCl, 1 NaH₂PO₄, 26.2 NaHCO₃, 1.3 MgSO₄, 2.5 CaCl₂, 10 D-glucose), and incubated for 30 min at 32 °C. Slices were then incubated for an additional 30 min at room temperature before recordings were performed. Slices were placed on an upright Zeiss Examiner D1 microscope equipped with a × 40 water-immersion objective in a recording chamber filled with carbogen-bubbled perfusion of ACSF containing 1 µM TTX and 0.1 mM picrotoxin, and maintained at 32 °C. Putative OLM neurons were identified by morphology, using a × 40 objective, as neurons with a large somata in the stratum oriens of the CA1 region of the hippocampus. These neurons were recorded from with patch pipettes with a resistance of 2.5–5 MΩ prepared from glass capillaries (Harvard Apparatus, cat. no. 300060, 1.5 mm OD, 0.86 mm ID) using a P-97 puller (Sutter Inst, Novato, CA). The intracellular pipette solution contained (in mM): 130 CsMeSO₃, 2.67 CsCl, 10 Hepes, 1 EGTA, 4 Mg-ATP, 0.3 Na-GTP, 3 QX-314 Cl, 5 TEA-Cl, 15 Creatine-phosphate disodium, 5 Creatinephosphokinase (mOsm = 303, pH = 7.44). Whole-cell patch-clamp recordings were obtained using a NPI ELC-03XS patch clamp amplifier and digitized using an InstruTECH ITC-18 data acquisition interface (HEKA Elektronik). A custom written procedure in Igor Pro 6.12A (provided by Oliver Schluter, ENI Goettingen) was used to visualize and store recorded data. Signals were low-pass filtered using a Bessel filter at 3 kHz and digitized at 10 kHz.

Whole-cell recordings were made for 2 min in basal conditions without capsaicin (12 sweeps of 10 s each), at a holding potential of − 75 mV to record mEPSCs, after which the perfusion solution was switched to ACSF containing 1 µM capsaicin. 50 additional sweeps (at 10 s intervals) were then recorded during capsaicin treatment. The input and series resistance (Rs) were monitored every 10 s during recording. Recordings where uncompensated Rs was > 30 MΩ or changed by more than ± 20% during recording were not used for analysis. Input resistances ranged from 100–400 MΩ and did not change by more than ± 20% during the course of the recording. Recorded baseline traces were analysed using Mini Analysis v6.0.3 software (Synaptosoft) with an amplitude threshold for detection of mEPSCs of approximately 3.5 × RMS noise. Average mEPSC frequency and amplitude were plotted with standard error of the mean. Cells in which the current deviated significantly from the standard deviation of the current recorded during a 2 min baseline prior to capsaicin addition, were considered responsive to capsaicin.

**Electrophysiological recording from hippocampal slices.** Acute hippocampal slices were prepared from 8–12 week old B6/J and TRPV1 KO adult male mice as previously described. Animals were anaesthetised with isoflurane and decapitated. The brain was removed from the skull and hippocampus was dissected. Transverse 400-µm-thick hippocampal slices were obtained using a tissue chopper (Stoelting). Slices were collected in ice-cold ACSF (124 mM NaCl, 4.9 mM KCl, 1.2 mM KH₂PO₄, 2.0 mM MgSO₄, 2.0 mM CaCl₂, 24.6 mM NaHCO₃, 10.0 mM D-glucose; saturated with 95% O₂ and 5% CO₂; pH 7.4 and 305 mOsm).

Hippocampal slices were incubated for 3 h at 32 °C and high oxygen tension (by bubbling with 95% O₂ and 5% CO₂; 30 l h⁻¹) in an interface chamber containing 2 ml ACSF. A constant flow (0.76 ml min⁻¹) of ACSF was perfused continuously during the incubation. For recording and stimulation of field

excitatory post-synaptic potentials (fEPSPs), monopolar lacquer-coated stainless steel electrodes (571000, A-M Systems) were positioned in the CA1 region. fEPSPs were recorded using a differential AC amplifier (Model 1700, A-M Systems) and a power (1401) analogue to digital converter (Cambridge Electronic Design) and tracked using the PWIN software (IFN Magdeburg). For each input, a determination of the test stimulation strength was stipulated as the current necessary to evoke a fEPSP of 40% of the maximal slope. 30 min after incubation with ACSF, the baseline recording was started with test stimuli consisting of 4 biphasic square pulses of constant current (0.2 Hz, 0.1 ms per polarity) every 5 min for at least 30 min. Long-term potentiation (LTP) was induced with one (1XTET) strong tetanization protocol consisting of 100 biphasic constant current pulses at 100 Hz and 0.2 ms per polarity. For LTP experiments testing the effect of nicotine, 5 µM nicotine was applied 30 min before LTP induction and throughout the recording (1 h).

**Statistical analysis**. All statistical analysis was carried out using Prism 6 software (GraphPad). All reported values in statistical analysis represent the mean of the indicated data set, and error bars indicate s.e.m. For pairwise comparisons and calculation of significance, Student's two-tailed type 2 $t$-tests with Welch's correction were performed, where $*P < 0.05$, $**P < 0.01$, and $***P < 0.001$. For multiple comparisons, significance was determined by one-way ANOVA with Tukey's *post hoc* test for multiple comparisons indicated by $*P < 0.05$, $**P < 0.01$, $***P < 0.001$, $****P < 0.0001$. For all statistical tests, data met the assumptions of the test, for example, normal distribution, and variance of individual groups was calculated and determined to be similar between groups that were statistically compared.

**Data availability**. The data that support the findings of this study are available within the article, its Supplementary Information files and from the corresponding author upon reasonable request.

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

## Acknowledgements

We thank Rory McQuiston, Julie Kauer, Manuela Schmidt, and Nils Brose for insightful discussions and critical review of the manuscript. This work was supported by NIH R44Da032464, R21DA033831, PA Cure 4100068719/Lehigh University Accelerator grants to J.M.M., and by a Sofja Kovalevskaja grant from the Alexander von Humboldt Foundation, European Research Council starting grant SytActivity FP7 260916, Deutsche Forschungsgemeinschaft grants CRC889, DE1951/1 and the Center for Nanoscale Microscopy and Molecular Physiology of the Brain (CNMPB), and the Boehringer Ingelheim Foundation, to C.D.

## Author contributions

J.I.H.-Z. conceived the research, designed, performed and analysed experiments, and co-wrote the paper. B.R., S.A. and R.H., designed, performed and analysed experiments. C.B., performed experiments and analysed the data. R.J.W. and K.A. performed experiments. M.A.S. designed and optimized experiments. A.A. analysed experiments. R.M.D., H.A.L. and J.M.M. designed experiments, and contributed animal models. J.F.S. and A.F. supervised and funded the work. C.D. conceived the research, designed and analysed experiments, supervised and funded the work and co-wrote the paper. All authors discussed the results and commented on the manuscript.

## Additional information

**Competing interests:** The authors declare no competing financial interests.

