## [Peer Review File · Nature Communications]

Reviewers' comments:

Reviewer #1 (Remarks to the Author):

The manuscript by Hurtado-Zavala et al. presents intriguing new information regarding TRPV1 channels in the hippocampus. The presence or absence of these channels has been controversial, and the authors go to some pains to use measures of mRNA, multiple antibodies, as well as Ca²⁺ imaging to convince the reader that a specific subtype of hippocampal interneurons, the OLM cells, indeed express this channel. Most interesting is the observation that in cultured neurons, 21 hr activation of the TRPV1 channels using the selective agonist, capsaicin, promotes an increased innervation by VGlut1 containing nerve terminals; this effect is absent in the *trpv1*^{-/-} mouse or when an antagonist was co-applied. The authors even express TRPV1 in cells not naturally expressing it, and find that a similar overnight treatment with capsaicin also recruits a larger number of excitatory nerve terminals.

The OLM interneurons have been shown to regulate LTP induction at excitatory CA3-CA1 pyramidal cell synapses, and have been proposed to mediate known effects of nicotine on this form of LTP. The authors of this paper go on to show that as previously shown, LTP is attenuated in the *trpv1*^{-/-} mouse, and that nicotine can rescue this defect; a selective beta2 subunit agonist also rescues LTP. This subunit is thought to be expressed in the OLM interneurons, suggesting that perhaps the rescue is a selective effect on the OLM cells. Overall the authors have made a good case for their arguments and the experiments are convincing.

Suggestions for revision:

1. It might be useful in resolving the controversy for the authors to suggest reasons why a previous group (Cavanaugh et al. 2013) did not find evidence using lacZ or PLAP reporter mice of TRPV1 present in hippocampal neurons.
2. The supplemental figure 2 is used as an argument that the N-terminal antibody the authors rely on is selective. However, this figure only shows a few examples, and also only addresses TRPV2 and TRPV4. What about TRPV3, which has been suggested to be present in hippocampus? Some quantification of these results would be much more useful than the single examples illustrating a small number of cells. Moreover, pictures and quantification in slices would be even more useful in trying to compare this work with the literature. Do the authors see staining in the dentate gyrus, as physiological effects have been reported here?
3. The finding of broadest interest to the field is the synaptogenic effect. While the authors localize TRPV1 to the postsynaptic OLM cell, the effects are presynaptic; an increase in excitatory nerve terminals and an increase in mEPSC frequency. It would be nice to test their hypothesis that NGF or BDNF are necessary for this result. Is the Ca²⁺ entry through TRPV1 channels the essential factor here? If field stimulation is substituted, is the same result obtained? Also one would presume that TRPV1 channels would rapidly desensitize in the continued presence of capsaicin (presumably the authors know how long the channel is activated from the Ca experiments); does the increased excitatory innervation occur with brief capsaicin activation?
4. In the discussion (p.10) the authors mention that *trpv1*^{-/-} mice would be expected to have increased temporoammonic LTP, as the OLM cells innervate this layer; they mention that the medial perforant path of dentate has increased LTP. However, this is a completely different pathway and is not presumably affected by OLM cells. The LTP here might be due to the LTD previously described at this and other synapses.
5. The authors favor the model that OLM neurons control the level of LTP at schaffer collaterals in CA1; however, the LTD described at these synapses is also absent in the knockouts, and may reasonably also contribute to minimizing schaffer collateral LTP. The recovery seen in the nicotinic agonists suggests a key role for the OLM cells, but while these data support the authors' model, they do not rule out other models that would incorporate the OLM cells and their interneuron targets.

Minor points

1. There is some confusion about the animals used for each experiment. E.g. for dissociated cultures, only the *trpv1* KOs are mentioned, while presumably littermates were also used.
2. The description of the slice physiology experiments was somewhat confusing. What are

“biphasic pulses at 0.1ms polarity”? What is the shape of the waveform? Also presumably the recording chamber was also an interface chamber?

Reviewer #2 (Remarks to the Author):

In this manuscript, the authors have found evidence for the selective expression of TRPV1 on OLM interneurons in the hippocampus, and that this appears to increase mainly excitatory synaptic innervation to these interneurons and play a role in synaptic plasticity. The authors have done a large amount of molecular and physiological experiments, and provided compelling evidence that TRPV1 is in fact expressed selectively and does appear to regulate excitatory input to the OLM interneurons. However there are two areas that are lacking in the present manuscript; how does expression of TRPV1 on these interneurons increase excitatory innervation, and what impact does this have behaviorally. The authors have discussed some possibilities on the former in the discussion, but have not presented any discussion on the later. For example has it been shown that the expression of TRPV1 on the OLM interneurons may regulate behaviors such as learning and memory. Such information could significantly increase the impact of the present manuscript.

Reviewer #3 (Remarks to the Author):

This study identifies a subpopulation of interneurons in the oriens-lacunosum moleculare (OLM) of the mouse hippocampus, which express functional TRPV1 channels and contribute to the total excitatory and inhibitory output of the hippocampus. The study combines immunofluorescence, pharmacology and electrophysiology techniques, among others.

The results are presented in a logical order and the methods described in detail, but the inappropriate statistical analysis used to analyse some data, the somehow confusing histograms and the way in which some immunofluorescence results are presented make the interpretation of the results hard. Also, the authors state that this is the first description of TRPV-1 presence in inhibitory interneurons in the hippocampus, since previous studies have localized them in excitatory neurons (lines 75-79). This statement is incorrect. Indeed, the recent paper from Lee and colleagues (J Neuroscience, 2015) that the authors cite describe the presence of TRPV1 on postsynaptic GABAergic synapses in the hippocampus, by means of immunogold staining for electromicroscopy. The novelty of the present paper is the characterization of the inhibitory subpopulation expressing TRPV1 channels.

Overall the data presented are interesting, but need some improvements. Electrophysiological data could be more relevant whether corroborated by pharmacological modulation of endogenous vanilloid agonist, like anandamide.

Specific comments:

- Figure 1F: to this reviewer it is not clear what the error bars on the x-axis refer to. The graph should represent the quantification of the fluorescence intensity of the two antibodies for TRPV-1 used in the study in neuronal cultures from WT and TRPV1 KO hippocampus. Therefore the statistics (the comparisons made) is not clear. Please, clarify.
- Figure 2B-C-D: neither the immunofluorescence panel nor the corresponding figure legend is clear. Without a proper labeling of the images it is hard to understand the quantitative analysis proposed in C. It should be better to indicate in each confocal image the marker used. Also, I suggest inverting the order between C and D. Finally, the quantification showed in the graph in C refers to seven markers, but the images are only six. Please provide the full panel of immunofluorescence staining.
- Figure 3. The statistical analysis used to compare multiple groups and conditions is inappropriate. One-way ANOVA should be used to compare for example TRPV1-expressing cells with/without pharmacological treatment.

- Figure 4. In the immunofluorescence depicted in A, the merge image of the KO control misses the somatostatin signal. Again in B, the statistical analysis is not appropriate for multiple group. Moreover, in D the only statistical significance reported in the graph is referred to TRPV-1 KO neurons, where reelin-somatostatin interneurons receive more vGAT inputs than the surrounding cells. The authors should refer to this data in the result section.
- Figure 6. As in figure 2B,C, the labeling of the figure and the corresponding graph should be revised.
- In general the statistics is poorly described and it should be useful to report mean \pm sem values in the text with number of animals/cells analyzed. Also, a specific paragraph for stistical analysis is missing in the method section.
- Some references are listed but not numbered in the text (for example Caterina et al, 2000, line 47).
- Several grammatical mistakes are present in the text.

Reviewer #4 (Remarks to the Author):

The authors of this manuscript address an interesting and important question about the function of TRPV1 channels in the hippocampus. They provide evidence that TRPV1 is expressed in OLM interneurons and that its expression is required for normal glutamatergic innervation of OLM interneurons in culture. They go on to report that TRPV1 knockouts exhibit impaired LTP that can be rescued by nicotine, a manipulation known to enhance recruitment of interneurons. These results lead to the proposition that TRPV1 is a synaptogenic factor for OLM interneurons that has an important role in gating synaptic Schaffer collateral plasticity through a previously described dis-inhibitory circuit. Overall this is an interesting topic with potential to be highly relevant for a broad audience. Identification of a synaptogenic function of TRPV1 seems novel. However, the link to circuit-level functions like LTP substantially reduce enthusiasm.

Major points

- 1). The main conclusion about the role of TRPV1 in plasticity is not well supported. Most, if not all, data relating to the function of TRPV1 in Ca²⁺ influx and excitatory innervation is performed in cell culture and the results are interpolated to a circuit-level phenomenon in slices. While the results in culture regarding the novel role of TRPV1 in innervation are mostly convincing, many additional experiments need to be performed in slices to relate those findings to SC plasticity. For example, the authors need to confirm functional expression of TRPV1 in OLM cells in WT slices, and then show that TRPV1 knockout reduces functional excitatory connectivity to OLM cells (in slices) in a manner that impairs their recruitment and therefore leads to dis-inhibition during LTP induction. Without these experiments to test the proposed mechanism, the conclusion based on the synapse number in culture (Figs 3-5) and LTP experiments in Fig 6 is premature.
- 2) The proposed mechanism involving low innervation of OLM cells in TRPV1 KOs implies a chronic function of TRPV1 in innervation (after long-term treatment, Figs 3-5) rather than an acute effect of TRPV1 channel activation (as been proposed by others). If the deficit in excitatory innervation of OLM neurons in the KOs explains the decrease in schaffer collateral LTP (as proposed on line 324), then acutely blocking or activating TRPV1 during LTP induction (with antagonists or agonists) should have no effect on LTP.

Minor Points

- 1) Some figures are poorly labeled, making it difficult to know what is being presented (i.e. Fig 1C, 2B).
- 2) Most of the data is presented as normalized values, but it is not always clear how or why the normalization was performed (i.e. 1F, 1H, 4B&D).
- 3) The authors should test whether stimulation and capsaicin evoked Ca²⁺ responses are blocked by a TRPV1 antagonist and glutamate receptor antagonists.
- 4) Including images of a surrounding (non-TRPV1) neurons and/or a KO culture would be useful in

the experiments showing how the number of synapses is increased by TRPV1 activation or decreased by TRPV1 blockade in culture.

5) Similarly, in Fig 4 all positive reelin-somatostatin cells are considered OLM neurons expressing TRPV1, but the concept of "surrounding cell" should be further clarified and included in the images.

6) In Fig 5, the authors should clarify the transfection efficiency and whether mouse or rat cultures were used (or results combined for both).

Reviewers' comments:

Reviewer #1 (Remarks to the Author):

The manuscript by Hurtado-Zavala et al. presents intriguing new information regarding TRPV1 channels in the hippocampus. The presence or absence of these channels has been controversial, and the authors go to some pains to use measures of mRNA, multiple antibodies, as well as Ca²⁺ imaging to convince the reader that a specific subtype of hippocampal interneurons, the OLM cells, indeed express this channel. Most interesting is the observation that in cultured neurons, 21 hr activation of the TRPV1 channels using the selective agonist, capsaicin, promotes an increased innervation by VGlut1 containing nerve terminals; this effect is absent in the *trpv1*^{-/-} mouse or when an antagonist was co-applied. The authors even express TRPV1 in cells not naturally expressing it, and find that a similar overnight treatment with capsaicin also recruits a larger number of excitatory nerve terminals.

The OLM interneurons have been shown to regulate LTP induction at excitatory CA3-CA1 pyramidal cell synapses, and have been proposed to mediate known effects of nicotine on this form of LTP. The authors of this paper go on to show that as previously shown, LTP is attenuated in the *trpv1*^{-/-} mouse, and that nicotine can rescue this defect; a selective beta2 subunit agonist also rescues LTP. This subunit is thought to be expressed in the OLM interneurons, suggesting that perhaps the rescue is a selective effect on the OLM cells. Overall the authors have made a good case for their arguments and the experiments are convincing.

Suggestions for revision:

1. It might be useful in resolving the controversy for the authors to suggest reasons why a previous group (Cavanaugh et al. 2013) did not find evidence using lacZ or PLAP reporter mice of TRPV1 present in hippocampal neurons.

Interestingly, Cavanaugh et al. did find a small population of non-pyramidal, reelin-expressing TRPV1-positive cells in the hippocampus in lacZ TRPV1 reporter mice (but not in PLAP TRPV1 reporter mice, which they say was likely due to greater sensitivity of nlacZ versus PLAP histochemistry). But Cavanaugh et al. did not detect TRPV1 expression in hippocampus using TRPV1 antisera or RT-PCR. We initially tried RT-PCR with the same primers used in Cavanaugh et al., and found similar results - we did not detect TRPV1 in adult brain. However, we were uncertain if this was due to the specific primers chosen or lack of detectable TRPV1. Therefore we designed three additional primer pairs for the TRPV1 pore region (also used to verify TRPV1 knockout, in which the pore region should be gone), the N-terminus, and the C-terminus of TRPV1. Using these primers, with TRPV1 knockouts as a control, we did detect TRPV1 in hippocampus by RT-PCR (Fig. 1A, B), and using TRPV1 antisera for Western blot (Fig. 1C) and immunocyto- and histochemistry (Fig.1E-I). We have added this information to the Discussion of the revised manuscript.

2. The supplemental figure 2 is used as an argument that the N-terminal antibody the authors rely on is selective. However, this figure only shows a few examples, and also only addresses TRPV2 and TRPV4. What about TRPV3, which has been suggested to be present in hippocampus? Some quantification of these results would be much more useful than the single examples illustrating a small number of cells.

We have quantified colocalization of TRPV1 with TRPV2 and TRPV4, and examined colocalization of TRPV1 with TRPV3 in revised Supplemental Figure 2B, C. Quantitation revealed that less than 3% of cells co-express TRPV1/TRPV2 or TRPV1/TRPV4, verifying that the TRPV1 antibody does not recognize TRPV2 or TRPV4. Interestingly, however, we found high co-localization of TRPV1 and TRPV3 (in 75% of cells). TRPV1 and TRPV3 knockouts (and not TRPV4 knockouts) have a similar reduction in Schaffer collateral LTP that is rescued by blocking GABAergic inhibition (Brown et al. Hippocampus 2013). In addition TRPV1 and TRPV3 proteins interact, and their co-expression enhances TRPV1 responses to capsaicin (Smith et al. Nature 2002). Thus it is possible these two isoforms cooperate in the same subset of inhibitory interneurons to affect LTP. To be sure that the TRPV3 antibody does not recognize TRPV1, we also immunostained TRPV1 over-expressing neurons with TRPV3 (new Supplemental Figure 1D); the TRPV3 antibody did not recognize over-expressed TRPV1, further verifying the specificity of both the TRPV3 and TRPV1 antibodies. This new information has been added to the revised manuscript.

Moreover, pictures and quantification in slices would be even more useful in trying to compare this

work with the literature. Do the authors see staining in the dentate gyrus, as physiological effects have been reported here?

We have added immunohistochemical analysis and quantitation of colocalization of TRPV1 with the putative VR.5' sv splice isoform, mGluR7, SOM, reelin, GAD65, VIP and PV in the stratum oriens of hippocampal sections in new Figure 2B, C (to complement the quantitation of these markers in dissociated hippocampal neurons - to which we have added repetitions in revised Figure 3D-F). We found similar results in stratum oriens of hippocampal sections compared to dissociated hippocampal neurons - TRPV1-expressing cells colocalize highly with VR.5' sv, mGluR7, SOM, reelin and GAD65, much less with PV, and not at all with VIP - consistent with TRPV1 being present in OLM neurons. There were more non-TRPV1-expressing reelin, GAD65 and VIP positive neurons in dissociated cultures than stratum oriens of hippocampal slices, most likely representing cells from other hippocampal regions. In stratum oriens sections, and repetitions of dissociated cultures, we detected a small population (20% of TRPV1-expressing cells in the stratum oriens) that were parvalbumin positive, consistent with a small subset of OLM neurons that have been reported to be parvalbumin-positive (Ferraguti et al. Hippocampus 2004, Chittajallu et al. Nat. Neurosci. 2013). This information has been added to the revised manuscript.

Regarding the question of expression of TRPV1 in the dentate gyrus, an example of an entire hippocampal slice, immunostained for TRPV1, including the dentate gyrus is shown below. We do see what looks like faint TRPV1 staining in specific cells in the hilus, but have not rigorously determined the number and type of TRPV1-expressing cells in this region by comparing immunostaining to knockouts (which is a bit more difficult in this case due to the slightly weaker TRPV1 antibody signal we have observed in this region). We therefore hesitate to comment on TRPV1 expression in the dentate gyrus in the current manuscript, but it is certainly an interesting topic for future studies.

3. The finding of broadest interest to the field is the synaptogenic effect. While the authors localize TRPV1 to the postsynaptic OLM cell, the effects are presynaptic; an increase in excitatory nerve terminals and an increase in mEPSC frequency. It would be nice to test their hypothesis that NGF or BDNF are necessary for this result. Is the Ca²⁺ entry through TRPV1 channels the essential factor here? If field stimulation is substituted, is the same result obtained? Also one would presume that TRPV1 channels would rapidly desensitize in the continued presence of capsaicin (presumably the authors know how long the channel is activated from the Ca experiments); does the increased excitatory innervation occur with brief capsaicin activation?

These are all interesting questions. To test if NGF, BDNF or Ca²⁺ entry through TRPV1 channels are necessary for the increase in excitatory synaptic innervation mediated by TRPV1, we blocked NGF (with bath application of 1 μg/ml TrkA-Fc), BDNF (with bath application of 0.4 μg/ml TrkB-Fc) and Ca²⁺ entry (by addition of 2 mM EGTA) during incubation with capsaicin for 21-24 hours, which when applied alone normally induces a significant increase in excitatory innervation, specifically in TRPV1-expressing neurons. Interestingly, in new Figure 8, we found that all of these treatments blocked the TRPV1-mediated synaptogenic effect. Capsaicin plus TrkA-Fc, or capsaicin plus TrkB-Fc, both reduced capsaicin-induced TRPV1-mediated excitatory innervation to control levels. Addition of 2 mM EGTA significantly reduced excitatory innervation below control levels, suggesting that calcium influx is necessary for the maintenance of synapses in general, and not just for synapse formation induced by TRPV1.

We also found that TRPV1-expressing neurons had increased BDNF levels compared to non-TRPV1-expressing neurons (new Figure 8A, B), and wild-type OLM cells had a greater number and intensity of BDNF puncta than TRPV1 knockout OLM cells (new Figure 8C, D). These results suggest a pathway whereby NGF upregulates TRPV1 in OLM neurons (former Supplemental Figure 3A-C, now main Figure 3D-F), which increases calcium influx and potentially neuronal activity. This activates BDNF, which promotes excitatory innervation of these cells. In support of this pathway, we found that overexpression of TRPV1 does not upregulate NGF (new Figure 8G) but does upregulate BDNF (new

Figure 8H), in transfected neurons (i.e. NGF upregulates TRPV1, which upregulates BDNF, rather than the other way around).

We also tested if increasing activity alone is sufficient to induce excitatory synapse formation on OLM neurons. To do this, we transduced hippocampal cultures with ChR2-EYFP AAV, or EGFP AAV as a control, and then delivered pulses of blue light via LED in the incubator for 24 hours. Although LED stimulation was sufficient to depolarize ChR2-EYFP expressing neurons (new Supplemental Figure 5A), there was no increase in excitatory innervation of TRPV1-expressing neurons that were transduced with ChR2-EYFP and stimulated, compared to EGFP controls (new Supplemental Figure 5B, C), suggesting that increased activity alone is not sufficient to induce excitatory innervation. However, in these experiments the majority of cells in the network in culture were transduced and the entire culture was stimulated. It is therefore still possible that specific differential levels of activity between cells may promote excitatory innervation in a Hebbian manner.

Finally, in new Supplemental Figure 4 we show that treatment with 50 nM or 1 μ M capsaicin for only 4 hours did not further increase excitatory innervation of TRPV1-expressing (on non-expressing) neurons, which is perhaps not surprising, given the time it takes for synapses to form and stabilize in culture. This also implies that channels either remain receptive to activation by capsaicin over long time periods, or cycle between active and desensitized states.

4. In the discussion (p.10) the authors mention that *trpv1*^{-/-} mice would be expected to have increased temporoammonic LTP, as the OLM cells innervate this layer; they mention that the medial perforant path of dentate has increased LTP. However, this is a completely different pathway and is not presumably affected by OLM cells. The LTP here might be due to the LTD previously described at this and other synapses.

We have modified this paragraph in the Discussion as: "In addition to reduced Schaffer collateral LTP, TRPV1 knockouts would therefore be expected to have increased temporoammonic LTP from the perforant path. Interestingly, TRPV1 knockouts do have increased LTP in the medial perforant pathway of the dentate gyrus (Chavez et al. 2010). However, the medial perforant pathway is presumably not affected by OLM cells. The increased LTP in this pathway in TRPV1 knockouts might rather be due to TRPV1-mediated LTD at the medial perforant path - dentate granular cell synapse (Chavez et al. 2010)."

5. The authors favor the model that OLM neurons control the level of LTP at schaffer collaterals in CA1; however, the LTD described at these synapses is also absent in the knockouts, and may reasonably also contribute to minimizing schaffer collateral LTP. The recovery seen in the nicotinic agonists suggests a key role for the OLM cells, but while these data support the authors' model, they do not rule out other models that would incorporate the OLM cells and their interneuron targets.

The "LTD" previously described in TRPV1 knockouts by Gibson et al. 2008, was long-term depression of stratum radiatum interneuron responses during high frequency stimulation of the Schaffer collateral pathway (not to be confused with "classical" LTD of excitatory connections between CA3 and CA1 pyramidal neurons in the Schaffer collateral pathway, induced by low frequency stimulation). Gibson et al. found a TRPV1-mediated depression of stratum radiatum interneuron responses during HFS of the Schaffer collateral which was absent in TRPV1 knockouts. This would result in "too much" inhibition of pyramidal neurons in TRPV1 knockouts, and reduce CA3 - CA1 Schaffer collateral LTP in TRPV1 knockouts (which we, and others have observed). Our model is largely in agreement with Gibson et al., except for where TRPV1 is located. We both agree that TRPV1 affects stratum radiatum interneurons, but is not present in stratum radiatum interneurons themselves (based on immunostaining in our case, and absence of blockade of depression by introducing TRPV1 blockers into recorded stratum radiatum interneurons in Gibson et al.). In terms of TRPV1 location, however, Gibson et al. proposed that TRPV1 is presynaptic on pyramidal neurons, where it reduces transmitter release and depresses post-synaptic stratum radiatum interneurons during HFS of the Schaffer collateral. We propose that TRPV1 is present in OLM neurons that contact stratum radiatum interneurons, where TRPV1 promotes excitatory innervation of OLM neurons. This would also act to depress stratum radiatum interneurons (via disinhibition) during HFS of the Schaffer collateral pathway. We favor this model because we observed capsaicin responses (in new Figure 6D, E) and TRPV1 staining specifically in OLM neurons (and not in TRPV1 knockouts), and because specific activation of OLM neurons by α 2 nicotinic receptors rescued LTP in TRPV1 knockouts. However, it remains possible that both pathways could work in parallel. We have added these points to the Discussion of the revised manuscript.

Minor points

1. There is some confusion about the animals used for each experiment. E.g. for dissociated cultures, only the *trpv1* KO mice are mentioned, while presumably littermates were also used.

Thank you. We have corrected this omission in the text.

2. The description of the slice physiology experiments was somewhat confusing. What are "biphasic pulses at 0.1ms polarity"? What is the shape of the waveform? Also presumably the recording chamber was also an interface chamber?

The Methods section "Electrophysiological recording from hippocampal slices" mentions that we used an interface chamber, and that biphasic pulses were delivered at 0.1 ms/polarity, i.e. 0.1 ms per polarity. Perhaps the forward slash was not in the final pdf, but hopefully it is present now. We have also added the waveform as, "...test stimuli consisting of 4 biphasic square pulses..."

Reviewer #2 (Remarks to the Author):

In this manuscript, the authors have found evidence for the selective expression of TRPV1 on OLM interneurons in the hippocampus, and that this appears to increase mainly excitatory synaptic innervation to these interneurons and play a role in synaptic plasticity. The authors have done a large amount of molecular and physiological experiments, and provided compelling evidence that TRPV1 is in fact expressed selectively and does appear to regulate excitatory input to the OLM interneurons. However there are two areas that are lacking in the present manuscript; how does expression of TRPV1 on these interneurons increase excitatory innervation, and what impact does this have behaviorally. The authors have discussed some possibilities on the former in the discussion, but have not presented any discussion on the later. For example has it been shown that the expression of TRPV1 on the OLM interneurons may regulate behaviors such as learning and memory. Such information could significantly increase the impact of the present manuscript.

In terms of how expression of TRPV1 might increase excitatory innervation, we have tested several possibilities (as described in response to Reviewer 1, point 3 above) in the revised manuscript. Based on this new data we propose a pathway whereby NGF upregulates TRPV1 in OLM neurons, which increases calcium influx. This activates BDNF, which then promotes excitatory innervation of these cells: In new Figure 8, we found that blockade of NGF (with bath application of 1 µg/ml TrkA-Fc), BDNF (with bath application of 0.4 µg/ml TrkB-Fc) or Ca²⁺ entry (by addition of 2 mM EGTA) during incubation with capsaicin for 21-24 hours, which normally induces excitatory innervation specifically in TRPV1-expressing neurons, all blocked the TRPV1-mediated synaptogenic effect. TRPV1-expressing neurons also had increased BDNF levels compared to non-TRPV1-expressing neurons (new Figure 8A, B), and wild-type OLM cells had a greater number and intensity of BDNF puncta than TRPV1 knockout OLM cells (new Figure 8C, D). These results suggest that NGF upregulates TRPV1 (former Supplemental Figure 3A-C, now main Figure 3D-F), which activates BDNF, which then promotes excitatory innervation. In support of this pathway, we found that overexpression of TRPV1 does not upregulate NGF (new Figure 8G) but does upregulate BDNF (new Figure 8H), in transfected neurons (i.e. NGF upregulates TRPV1, which upregulates BDNF, rather than the other way around). We also found, by stimulating ChR2-EYFP AAV transduced neurons with pulses of blue light via LED in the incubator for 24 hours, that increased activity alone does not increase excitatory innervation of TRPV1-expressing neurons (new Supplemental Figure 5).

In terms of behavior, TRPV1 knockout mice exhibit less anxiety than wild-type mice, and have reduced hippocampal-dependent conditioned and contextual fear memory compared to wild-type mice (Marsch et al. 2007). These behavioral deficits in memory are consistent with a reduction of hippocampal processing and the reduced LTP observed in the Schaffer collateral pathway of TRPV1 knockout mice. We have added this information to the Discussion of the revised manuscript, following our previous short comment on behavior: "A subset of OLM cells with high excitation are recruited during hippocampal sharp wave ripple events⁶⁸. These events are important for memory consolidation and reactivation of place cells⁶⁹. TRPV1 may be involved in sharp wave ripples - by promoting high excitatory innervation of this subset of OLM neurons - and therefore important for intrahippocampal information processing necessary for spatial memory."

We hesitate to speculate further on specific behavioral impacts of TRPV1 or OLM neurons, but it would certainly be very interesting to compare TRPV1 knockouts to mice in which OLM neurons are optogenetically silenced, for example.

Reviewer #3 (Remarks to the Author):

This study identifies a subpopulation of interneurons in the oriens-lacunosum moleculare (OLM) of the mouse hippocampus, which express functional TRPV1 channels and contribute to the total excitatory and inhibitory output of the hippocampus. The study combines immunofluorescence, pharmacology and electrophysiology techniques, among others.

The results are presented in a logical order and the methods described in detail, but the inappropriate statistical analysis used to analyse some data, the somehow confusing histograms and the way in which some immunofluorescence results are presented make the interpretation of the results hard. Also, the authors state that this is the first description of TRPV1 presence in inhibitory interneurons in the hippocampus, since previous studies have localized them in excitatory neurons (lines 75-79). This statement is incorrect. Indeed, the recent paper from Lee and colleagues (J Neuroscience, 2015) that the authors cite describe the presence of TRPV1 on postsynaptic GABAergic synapses in the hippocampus, by means of immunogold staining for electromicroscopy. The novelty of the present paper is the characterization of the inhibitory subpopulation expressing TRPV1 channels.

Thank you very much for catching this error. Lee et al., J. Neurosci. 2015 did indeed observe post-synaptic TRPV1 at symmetric (i.e. inhibitory-inhibitory) synapses by immunogold. We have revised the Introduction to read: "...Most previous studies describing a function of TRPV1 in synaptic plasticity have localized TRPV1 to excitatory neurons^{7, 9, 15}, which comprise the principle cells of the glutamatergic tri-synaptic circuit in the hippocampus. However, TRPV1 has also recently been reported at post-synaptic sites in inhibitory neurons¹⁷ (Lee et al. 2015)...". We have also removed the Lee et al. 2015 reference from the section of the Discussion describing previous studies that have localized TRPV1 to excitatory neurons. Regarding the former point about analysis and data presentation, responses to specific issues are outlined below.

Overall the data presented are interesting, but need some improvements. Electrophysiological data could be more relevant whether corroborated by pharmacological modulation of endogenous vanilloid agonist, like anandamide.

We assume that the reviewer is interested in which endogenous ligands of TRPV1 might lead to the enhanced excitatory innervation of OLM neurons. This is an interesting and important question we are also interested in, but is difficult to test. Ideally one would block the synthesis or action of specific endogenous ligands to determine if this blocks the synaptogenic effect mediated by TRPV1. However, since the effect of capsaicin is not acute, but requires long-term treatment to induce increased TRPV1-mediated excitatory innervation of OLM cells, we assume that endogenous agonists would also require long time periods to exert their effects, and could do so at any time during development, which is difficult to assay. In addition, many agonists of TRPV1, like anandamide and 2-AG, can activate other receptors (i.e. the CB1 cannabinoid receptor), the blockade of which would affect processes mediated by receptors other than TRPV1 and could confound interpretation. There are also a large number of endogenous TRPV1 ligands (e.g. 12-HPETE, anandamide, N-arachidonoyl-dopamine (NADA), 2-arachidonylglycerol (2-AG) and N-oleoylethanolamide) which are present in the brain, and (as mentioned in the Discussion) it is possible that different agonists promote excitatory innervation of OLM cells at different times, or in response to distinct stimuli. Thus, while it is an interesting question, unfortunately it is currently beyond our capabilities to investigate in a rigorous manner.

Specific comments:

- Figure 1F: to this reviewer it is not clear what the error bars on the x-axis refer to. The graph should represent the quantification of the fluorescence intensity of the two antibodies for TRPV1 used in the study in neuronal cultures from WT and TRPV1 KO hippocampus. Therefore the statistics (the comparisons made) is not clear. Please, clarify.

For immunostaining experiments, we always added an internal control of "black level" in surrounding non-TRPV1-expressing neurons in each experiment, to be as rigorous as possible, given that some amount of TRPV1-positive signal for the C-terminal antibody remained in the TRPV1 knockouts. However, because this is a bit confusing in the figure itself, we have removed the "black level" error bars on the x-axis and have described how quantitation was done in more detail in the "Somatic protein expression quantitation" section of the Methods.

- Figure 2B-C-D: neither the immunofluorescence panel nor the corresponding figure legend is clear. Without a proper labeling of the images it is hard to understand the quantitative analysis proposed in C. It should be better to indicate in each confocal image the marker used. Also, I suggest inverting the

order between C and D. Finally, the quantification showed in the graph in C refers to seven markers, but the images are only six. Please provide the full panel of immunofluorescence staining.

Ack! It appears that the image labels for Figure 2B were lost in the pdf. Our apologies for not noticing this at submission. These images are now properly labelled (in Figure 3A, B, C in the revised manuscript). We have also inverted the order of original panels 2C and D (now panels 3B and C), such that mGluR7 immunostaining is shown first, followed by quantitation of all markers, as suggested. The quantitation includes seven markers, where the immunostain of VR.5' sv (the splice isoform of TRPV1 that remains in the TRPV1 knockouts and is recognized by the C-terminal TRPV1 antibody) relative to full length TRPV1 (labelled with the N-terminal TRPV1 antibody), is shown in Figure 1E where the presence of the splice isoform in hippocampus is initially described. This is now referred to in the text of the revised manuscript.

- Figure 3. The statistical analysis used to compare multiple groups and conditions is inappropriate. One-way ANOVA should be used to compare for example TRPV1-expressing cells with/without pharmacological treatment.

We have re-analyzed the data presented in former Figure 3 (now Figure 4 in the revised manuscript) assessing excitatory and inhibitory synapse formation on TRPV1-expressing hippocampal neurons, using one-way ANOVA with Tukey's post hoc test for multiple comparisons to determine significance and generate p values for pair-wise comparisons within the data set. This correction showed that the minor changes (in vGAT innervation, for example) are not significant, but our main findings - that vGluT1 innervation of OLM neurons is increased by TRPV1, further increased by capsaicin, and blocked by the TRPV1 channel antagonist SB-366791 and in TRPV1 knockouts - remain highly significant. We have revised information regarding the statistical analysis in the new Figure 4 Legend, and added a paragraph in the Methods section of the revised manuscript describing Statistical Analysis.

- Figure 4. In the immunofluorescence depicted in A, the merge image of the KO control misses the somatostatin signal. Again in B, the statistical analysis is not appropriate for multiple group. Moreover, in D the only statistical significance reported in the graph is referred to TRPV-1 KO neurons, where reelin-somatostatin interneurons receive more vGAT inputs than the surrounding cells. The authors should refer to this data in the result section.

In Figure 4 (now Figure 5 in the revised manuscript) the merge image indicates a combination of vGluT, reelin and MAP-2 (in panel A), or vGAT, reelin and MAP-2 (in panel C). This has now been clarified in the revised figure legend. Technically, overlap of red, green and blue yields white, so more than four channels cannot be merged (while still conveying information from each channel). Panel C appears white in cell bodies, because vGAT, reelin and MAP-2 overlap in this region (the vGAT antibody also stains the cell body of inhibitory neurons).

In addition, we have re-analyzed the data in former Figure 4 (now Figure 5 in the revised manuscript) assessing excitatory and inhibitory synapse formation on OLM neurons, using one-way ANOVA with Tukey's post hoc test for multiple comparisons, as mentioned above. Minor changes (in vGAT innervation, for example) are no longer significant, but our main findings - that vGluT1 innervation of OLM neurons is increased by TRPV1, further increased by capsaicin, and absent in TRPV1 knockouts - remain highly significant. We have revised the statistical analysis in the new Figure 5 Legend, and added a Statistical Analysis paragraph to the Methods section.

- Figure 6. As in figure 2B,C, the labeling of the figure and the corresponding graph should be revised.
- In general the statistics is poorly described and it should be useful to report mean±sem values in the text with number of animals/cells analyzed. Also, a specific paragraph for stistical analysis is missing in the method section.

We have revised Figure 6 (now Figure 9 in the revised manuscript) to include an image of a hippocampal section from a CHRN2-EGFP mouse immunostained with TRPV1 and reelin (which was originally included in the quantitation, but not shown as an image).

In addition we have revised the manuscript to describe the statistics in greater detail including the number of animals/ cells analyzed (currently listed in the Figure Legends). The text may become difficult to read if we add all mean±sem numbers to the text itself for each graph shown, and this does not seem to be standard format for many publications in Nature Communications as far as we can tell, so we have not added this information to the text for the moment. If, however, the reviewer and editor deem this information necessary in the text itself, we can add numbers for mean±sem. We have,

however, added a paragraph on Statistical Analysis to the Methods section, as suggested.

- Some references are listed but not numbered in the text (for example Caterina et al, 2000, line 47).

Thank you. The incorrectly formatted references, including Caterina et al., 2000, have been corrected.

- Several grammatical mistakes are present in the text.

Thank you. We have read through the text carefully and corrected mistakes (but if specific additional issues remain we would be happy to correct them).

Reviewer #4 (Remarks to the Author):

The authors of this manuscript address an interesting and important question about the function of TRPV1 channels in the hippocampus. They provide evidence that TRPV1 is expressed in OLM interneurons and that its expression is required for normal glutamatergic innervation of OLM interneurons in culture. They go on to report that TRPV1 knockouts exhibit impaired LTP that can be rescued by nicotine, a manipulation known to enhance recruitment of interneurons. These results lead to the proposition that TRPV1 is a synaptogenic factor for OLM interneurons that has an important role in gating synaptic Schaffer collateral plasticity through a previously described dis-inhibitory circuit. Overall this is an interesting topic with potential to be highly relevant for a broad audience. Identification of a synaptogenic function of TRPV1 seems novel. However, the link to circuit-level functions like LTP substantially reduce enthusiasm.

Major points

1). The main conclusion about the role of TRPV1 in plasticity is not well supported. Most, if not all, data relating to the function of TRPV1 in Ca²⁺ influx and excitatory innervation is performed in cell culture and the results are interpolated to a circuit-level phenomenon in slices. While the results in culture regarding the novel role of TRPV1 in innervation are mostly convincing, many additional experiments need to be performed in slices to relate those findings to SC plasticity. For example, the authors need to confirm functional expression of TRPV1 in OLM cells in WT slices, and then show that TRPV1 knockout reduces functional excitatory connectivity to OLM cells (in slices) in a manner that impairs their recruitment and therefore leads to dis-inhibition during LTP induction. Without these experiments to test the proposed mechanism, the conclusion based on the synapse number in culture (Figs 3-5) and LTP experiments in Fig 6 is premature.

This is an important point. To test our findings electrophysiologically, we recorded from OLM neurons in hippocampal slices from wild-type and TRPV1 knockout mice. Because OLM neurons represent the vast majority of inhibitory interneurons in the stratum oriens of hippocampal slices (Pouille & Scanziani 2004, McBain et al. 1994), we identified putative OLM cells in this region morphologically (as cells with large somata in the stratum oriens near the alveus) for whole cell patch clamp experiments. In new Figure 6, we found that mEPSC frequency was significantly reduced in putative OLM neurons in TRPV1 knockout hippocampal slices compared to those in wild-type slices, while mEPSC amplitude was unchanged. To confirm that these neurons expressed TRPV1, we also recorded currents in putative OLM cells in response to perfusion of 1 μ M capsaicin in wild-type and TRPV1 knockout hippocampal slices. In wild-type slices, 9 out of 12 neurons responded to capsaicin (as shown in new Figure 6D sample traces). In 19 recordings from TRPV1 knockout slices, we did not see responses to capsaicin (new Figure 6E), except for one 200 pA apparent hyperpolarizing response in one cell, and another 400 pA depolarizing response in another, which were determined to be outliers by ROUT analysis of current peak amplitude. These data confirm our findings that TRPV1 promotes excitatory innervation of OLM neurons, and that TRPV1 knockout reduces functional excitatory connectivity to OLM cells in slices, which could lead to reduced disinhibition and therefore reduced LTP.

2) The proposed mechanism involving low innervation of OLM cells in TRPV1 KOs implies a chronic function of TRPV1 in innervation (after long-term treatment, Figs 3-5) rather than an acute effect of TRPV1 channel activation (as been proposed by others). If the deficit in excitatory innervation of OLM neurons in the KOs explains the decrease in schaffer collateral LTP (as proposed on line 324), then acutely blocking or activating TRPV1 during LTP induction (with antagonists or agonists) should have no effect on LTP.

To test this we performed LTP experiments in wild-type slices in which we acutely added 1 μ M capsaicin or 1 μ M SB-366791 to activate or block TRPV1 channels, respectively. We found no

significant effects of capsaicin or SB-366791 on LTP compared to control hippocampal slices (in new Supplemental Figure 6), consistent with our hypothesis that TRPV1-mediated effects on LTP result from innervation of OLM cells, rather than acute TRPV1 activation during LTP. Previous studies also report no effect of SB-366791 (Li et al. 2008), or the TRPV1 antagonist capsazepine (Bennion et al. 2011) on LTP. However, these studies do report an increase in LTP with addition of capsaicin (Li et al., 2008, Bennion et al. 2011); a 1000-fold higher concentration of capsaicin was used in the former publication, and a theta burst protocol was used in the latter. It is therefore likely that the concentration of capsaicin or induction protocol may affect the degree of potentiation of LTP by capsaicin. In any case, a capsaicin-induced increase in LTP is also consistent with our model - where capsaicin (like nicotine) would activate TRPV1 in OLM neurons, promote disinhibition of the Schaffer collateral pathway, and increase LTP, irrespective of changes in innervation of OLM neurons.

Minor Points

1) Some figures are poorly labeled, making it difficult to know what is being presented (i.e. Fig 1C, 2B).

As mentioned in response to Reviewer 3 it appears that the image labels for Figure 2B were lost in the pdf. We apologize for this oversight. These images are now properly labelled (in Figure 3A, B, C in the revised manuscript). We have also added "WT KO" to the lower blot of Figure 1C, to make it clear that the lanes are the same as those depicted in the upper blot, and clarified in the text that the upper blot shows TRPV1 in whole brain and the lower blot shows TRPV1 in hippocampus.

2) Most of the data is presented as normalized values, but it is not always clear how or why the normalization was performed (i.e. 1F, 1H, 4B&D).

As also mentioned above in response to Reviewer 3, regarding Fig. 1F, immunostaining experiments were internally controlled by setting the "black" level" to surrounding non-TRPV1-expressing neurons, given that TRPV1 signal remained in the TRPV1 knockouts, where TRPV1 intensity was normalized to black levels. Because this is confusing in the figure itself, we have removed the "black level" error bars on the x-axis and instead have described the quantitation in the Methods. We have also therefore removed the "norm" descriptor in the y-axis of Fig. 1F and 1H. In Figure 4B & D (now Figure 5B & D), vGluT and vGAT synapse number was normalized to that on surrounding non-TRPV1-expressing neurons for comparison. This information has been added to the Figure Legend.

3) The authors should test whether stimulation and capsaicin evoked Ca²⁺ responses are blocked by a TRPV1 antagonist and glutamate receptor antagonists.

We tested this by repeating calcium imaging experiments in the presence of 10 μ M APV, 10 μ M CNQX, and 1 μ M TTX, in dissociated rat hippocampal cultures treated with 1 μ M capsaicin, compared to cultures containing 1 μ M SB-366791 (to block TRPV1) and treated with 1 μ M capsaicin. New Supplemental Figure 1 shows that the capsaicin-induced calcium influx remained in a sub-population of neurons (23 of 574 cells) in the presence of TTX and CNQX, and treatment of cultures with the TRPV1 antagonist SB-366791 blocked this capsaicin-induced calcium influx (only 1 of 545 cells showed a slight response during capsaicin addition). This further confirms that the capsaicin response is due to TRPV1 expressed in a sub-population of hippocampal neurons.

4) Including images of a surrounding (non-TRPV1) neurons and/or a KO culture would be useful in the experiments showing how the number of synapses is increased by TRPV1 activation or decreased by TRPV1 blockade in culture.

In new Supplemental Figure 4A we have included larger field images of wild-type and TRPV1 knockout neurons that include both TRPV1-expressing and non-expressing surrounding neurons following capsaicin treatment, so that the number of vGluT1-positive synapses can be directly compared.

5) Similarly, in Fig 4 all positive reelin-somatostatin cells are considered OLM neurons expressing TRPV1, but the concept of "surrounding cell" should be further clarified and included in the images.

We have included larger field images of wild-type and TRPV1 knockout neurons including both reelin/somatostatin-expressing and non-expressing surrounding neurons following capsaicin treatment, so that vGluT1-positive synapses can be directly compared, in new Supplemental Figure 4D.

6) In Fig 5, the authors should clarify the transfection efficiency and whether mouse or rat cultures were used (or results combined for both).

In former Figure 5 (now Figure 7 in the revised manuscript), rat cultures were used, and neurons were transfected with Lipofectamine 2000, which results in a relatively low (1-5%) efficiency of transfection. Thus virtually all presynaptic terminals contacting TRPV1 (or EGFP control) transfected cells would be expected to be "wild-type", where TRPV1 is only overexpressed post-synaptically. We can therefore attribute any changes in presynaptic innervation or strength to post-synaptic TRPV1. This information has been added to the revised manuscript.

REVIEWERS' COMMENTS:

Reviewer #1 (Remarks to the Author):

The authors have addressed my comments and I am satisfied with their response to other reviewers as well.

Reviewer #2 (Remarks to the Author):

Authors have satisfactorily addressed my concerns.

Reviewer #3 (Remarks to the Author):

All criticisms have been overcome in this revision, and I have no further comments to the authors. The manuscript could be suitable for publication in its present form.